# Oil palm modelling in the global land-surface model ORCHIDEE-MICT

Yidi Xu[1], Philippe Ciais[2], Le Yu[1,3*], Wei Li[1*], Xiuzhi Chen[4], Haicheng Zhang[5], Chao Yue[6], Kasturi Kanniah[7], Arthur P. Cracknell[8], Peng Gong[1,3]

[1]Ministry of Education Key Laboratory for Earth System Modeling, Department of Earth System Science, Tsinghua University, Beijing, 100084, China

[2]Laboratoire des Sciences du Climat et de l'Environnement, LSCE/IPSL, CEA-CNRS-UVSQ, Universite Paris-Saclay, Gif-sur-Yvette 91191, France

[3]Joint Center for Global Change Studies, Beijing 100875, China

[4]Guangdong Province Key Laboratory for Climate Change and Natural Disaster Studies, School of Atmospheric Sciences, Sun Yat-sen University, Guangzhou 510275, China

[5]Department Geoscience, Environment and Society, Université Libre de Bruxelles, 1050 Bruxelles, Belgium

[6]State Key Laboratory of Soil Erosion and Dryland Farming on the Loess Plateau, Northwest A&F University, Yangling, Shaanxi 712100, China

[7] Centre for Environmental Sustainability and Water Security (IPASA), Research Institute for Sustainable Environment (RISE) and Tropical Map Research Group, Faculty of Built Environment and Surveying, Universiti Teknologi Malaysia, Johor Bahru, Johor, 81310, Malaysia

*Correspondence to*: Le Yu (leyu@tsinghua.edu.cn) and Wei Li (wli2019@tsinghua.edu.cn)

**Abstract.** Oil palm is the most productive oil crop that provides ~40% of the global vegetable oil supply, with 7% of the cultivated land devoted to oil plants. The rapid expansion of oil palm cultivation is seen as one of the major causes for deforestation emissions and threatens the conservation of rain forest and swamp areas and their associated ecosystem services in tropical areas. Given the importance of oil palm in oil production and its adverse environmental consequences, it is important to understand the physiological and phenological processes of oil palm and its impacts on the carbon, water and energy cycles. In most global vegetation models, oil palm is represented by generic plant functional types (PFT) without specific representation of its morphological, physical and physiological traits. This would cause biases in the subsequent simulations. In this study, we introduced a new specific PFT for oil palm in the global land surface model ORCHIDEE-MICT (v8.4.2). The specific morphology, phenology and harvest process of oil palm were implemented, and the plant carbon allocation scheme was modified to support the growth of branch and fruit component of each phytomer. A new age-specific parameterization scheme for photosynthesis, autotrophic respiration, and carbon allocation was also developed for the oil palm PFT, based on observed physiology, and was calibrated by observations. The improved model generally reproduces the leaf area index, biomass density and fruit yield during the life cycle at 14 observation sites. Photosynthesis, carbon allocation and biomass components for oil palm also agree well with observations. This explicit representation of oil palm in global land surface model offers a useful tool for understanding the ecological processes of oil palm growth and assessing the environmental impacts of oil palm plantations.

# 1 Introduction

Oil palm is one of the most important vegetative oil crops in the world. It provides 39% of the global supply of vegetable oil and occupies 7% of the agricultural land devoted to oil-producing plants (Caliman, 2011; Rival and Levang, 2014). With the increasing demand for palm oil as a biofuel and a feedstock for industrial products, oil palm plantation continuously expanded from 5.59 to 19.50 million ha during 2001-2016 in the world's top two palm oil producers, Malaysia and Indonesia (Xu et al., 2020). This rapid expansion brought about high ecological and social costs. About half of the oil palm cultivation lands were converted from biodiverse tropical forests during 1990-2005 (Koh and Wilcove, 2008), leading to losses of habitats (Fitzherbert et al., 2008), peatlands (Koh et al., 2011; Miettinen et al., 2016) and carbon emissions from land use change (Guillaume et al., 2018). Land use change (LUC) from peat swamp forest to oil palm plantation contributed about 16-28% of the total national greenhouse gas (GHG) emissions in Southeastern Asia (Cooper et al., 2020). A comprehensive understanding of fruit production, land use change, carbon emissions and other environmental consequences of oil palm is urgently needed for guiding more sustainable management practices.

Many field-based studies underpinned the specific phenology and growth of oil palm and its key physiological processes (Noor and Harun, 2004; Lamade and Bouillet, 2005; Sunaryathy et al., 2015: Ahongshangbam et al., 2019). Models developed based on these field observations provide a useful tool for large-scale simulation of oil palm growth and yields and their impacts on the regional carbon, water and energy budgets. Oil palm growth models have been developed to simulate the biomass yields of oil palm based on the physiological processes and phenological characteristics such as flowering and rotation dynamics (Van Kraalingen et al., 1989; Henson, 2009; Combres et al., 2013; Hoffmann et al., 2014; Huth et al., 2014; Paterson et al., 2015; Teh and Cheah 2018). Although these models can generally reproduce the observed yields, they are usually applied for fruit production simulation without the whole carbon, water and energy cycle, do not allow the representation of land-use changes, thus usually cannot be integrated for regional and global gridded simulations like land surface models.

Alternatively, process-based land surface models (LSMs) can simulate spatially explicit plant growth, biomass density and yield and a full set of carbon, nutrient, water and energy fluxes and storage pools (Fisher et al., 2014). Vegetation in most LSMs is represented by a discrete number of plant functional types (PFTs) and oil palm is approximated by tropical broadleaved evergreen (TBE) trees without a specific representation in LSMs (except CLM-Palm), although the physiological characteristics of oil palm differ from generic TBE trees. For example, the maximum leaf area index (LAI) of oil palm is up to 6 $m^2$ $m^{-2}$ depending on the genotypes and locations, which is lower than TBE (8 $m^2$ $m^{-2}$) in Indonesia and other plantations such as rubber (9 $m^2$ $m^{-2}$) (Vernimmen et al., 2007; Propastin, 2009; Rusli and Majid, 2014). The Maximum rate of carboxylation, $V_{cmax25}$ of mature oil palm, by contrast, is higher than in natural tropical forests (Carswell et al., 2000; Kattge et al., 2009; Teh Boon Sung and See Siang, 2018). Oil palm has a shallower rooting system and lower above ground biomass compared to forests (Carr, 2011), and its above and below ground biomass ratio is lower than the natural forests (Kotowska et al., 2015). To maintain a huge fruit productivity with shallow roots, a large amount of water is required by oil palm for evapotranspiration (~4-6 mm $d^{-1}$), typically 25% higher than in tropical forests in the same region (Meijide et al., 2017; Manoli

et al., 2018). Ignoring those differences in the parameterizations of LSMs would cause biases when simulating oil palm growth, yields and the biophysical processes in a large-scale model application, which calls for new parameterizations dedicated to oil palm as a specific PFT in those models.

Oil palm has a specific morphology, phenology and management practice compared to other perennial crops and tropical evergreen forests. Oil palm has a solitary columnar stem with phytomers (palm branches supporting leaves and fruit bunches) produced in succession at the top of stem. Fruit bunches are developed in the axil of each phytomer and each phytomer experiences a life cycle from leaf initiation, inflorescences and fruit developing to harvest and pruning (Corley and Tinker, 2015; Lewis et al., 2020). At the maturity stage, one oil palm tree holds ~40 visible expanded phytomers from the youngest to the oldest, and 40-60 initiating phytomers within the apical buds (Combres et al., 2013). It takes about 2-3 years for the reproductive organ to develop before flower initiation and fruit harvest (Corley and Tinker, 2015). Currently, the biomass pool of phytomers is not included in the generic tree PFTs of most land surface models (except CLM-Palm), which prevents us from modelling phytomer-specific development, monthly harvest and pruning. In addition, the closest PFT of oil palm in the model, known as TBE, has a different leaf phenology —with a higher old leaf turnover and increased new leaf production in the dry season, based on the satellite and ground based observations (Wu et al., 2016). This leaf phenology scheme was parameterized for leaf age cohorts in ORCHIDEE (one of commonly used LSMs) for Amazonian evergreen forest (Chen et al., 2020) but whether it can be adapted to the oil palm or not needs further investigations. At the productive stage, regular harvest and pruning are applied to maintain the optimal number of phytomer and maximize harvested yields. Also, oil palm planted in mineral soil is managed in a rotation cycle of 25-30 years (manually cut) due to the difficulties in harvesting and the potential decline of fruit production (Hoffmann et al., 2014; Röll et al., 2015). Thus, oil palm cannot be described as an annual crop, neither as a natural tree PFT with a longevity of decades to centuries. Therefore, including forest age dynamics (Yue et al., 2018) is needed in a LSM to represent the management practice and cycle of growth, fruit harvest and rotation of oil palm at different age stages. CLM-Palm was the first LSM that introduced oil palm specific PFT and a sub-canopy/sub-PFT framework for modelling oil palm's phytomer-based structure and phenological and physiological traits in CLM4.5 (Fan et al., 2015). This work provides an important conceptual framework for implementing oil palm modelling in other LSMs.

In this study, we aimed to model oil palm growth from young to mature plants and the specific morphology, phenology and management characteristics in the ORCHIDEE LSM. Incorporating an oil palm PFT into ORCHIDEE would contribute to modeling the carbon, water and energy cycle of this perennial crop in a variety of LSMs except for CLM that already implemented oil palm modelling. The oil palm integration was based on existing leaf age cohorts-based phenology of TBE and distinct age classes of the model, but significant modifications have been made to accommodate the phenology, physiological and management characteristics of oil palm. The oil palm growth from leaf initiation, fruit development, maturity and to the clear-cutting of oil palm PFT at rotation were represented in the ORCHIDEE LSM. A sub-PFT structure—phytomer with branch and fruit (leaf component was implemented at PFT-level with four leaf age cohorts) for oil palm was implemented in ORCHIDEE based on the sub-PFT structure incorporated in the CLM-Palm (Fan et al., 2015). The plant carbon allocation

scheme was modified to support the growth of branch and fruit component of each phytomer. Management practice of pruning, fruit harvest and rotation were also implemented. The objectives of this study are to 1) implement growth (especially phytomer development), phenology and harvest processes for oil palm as a new PFT of the ORCHIDEE LSM, 2) adjust physiological and phenological parameters using field measurements, and 3) evaluate simulated biomass and oil palm yields at a range of sites across Indonesia, Malaysia and Benin.

## 2 Model development and parameterization

### 2.1 Observation Data

Data from 14 sites with reported coordinates were collected from published literatures for model validation (Table S1). Since tropical humid climate is favourable for oil palm growth, most of in situ measurements are located in Indonesia (6 sites) and Malaysia (7 sites) except one site in Benin (Figure 1). The observation sites have high mean annual precipitation (MAP, 574.2-3598.8 mm yr$^{-1}$) and high mean annual temperatures (MAT) between 24.3°C and 28.8°C throughout the year, which covers 97.27% and 85.14% range of MAP and MAT in the global oil palm plantation area respectively in 2010 (Cheng et al., 2018) (Figure S1). The MAT, MAP and clay fraction (CF) for the global oil palm plantation area were based on the climate data from the CRUNCEP gridded dataset (Viovy, 2011) and the Harmonized World Soil Database (HWSD v1.2, (Nachtergaele et al., 2010)). The observation sites include 4 smaller plantations (<50 ha, Site 1 and 2 for smallholders and Site 4, 5 and 12 for research sites, Figure 1) and 9 industrial plantations up to 23625 ha. Site 12 and Site 14 were covered by very deep peat soil before oil palm cultivation, where the former natural vegetation was peat swamp forest. The natural vegetation in other sites was dominated by tropical rainforest and clay fraction varies from 0-11% (Figure S1). LAI, gross primary productivity (GPP), net primary productivity (NPP), fruit bunches (yield) and biomass at different ages including young and mature oil palms were collected from these sites for model validation. Annual data of total biomass and yields were available for Site 3 and Site 12. The biomass data at Site 3 was calculated by allometric equation using the measured diameter at breast height (DBH) and height of the stem (Corley and Tinker 2015), while yield data at Site 12 was obtained from measurements of harvested fruit bunch every time. Sites 1, 2, 12 and 13 provide observations of different NPP components by quantifying all the plant pools change for a specified time interval. Fractions of different biomass parts were collected by combining measurements of biomass partition and calculations using empirical equations in Site 12 and Site 3 (see details in Table S1). Due to the lack of accessible continuous observations in one or two sites, we have to utilize the existing knowledge of oil palm growth phenology and plantation management, together with the range of field observations from all the sites to constrain the model. We also added a test by recalibrating the model using data from Site 12 with more observations compared to other sites, and we then validated the model using data at the remaining sites (Figure S4 and S5). Facing the difficulty in acquiring the original harvest records for independent sites, we also ran simulations in the same site as previous studies (Figure 11 in Teh and Cheah 2018 and Figure 6 in Fan et al., 2015) and visually compare the temporal dynamics of simulated yields.

## 2.2 Model description

Organising Carbon and Hydrology in Dynamic Ecosystems (ORCHIDEE) is the land surface component of the French Institut Pierre Simon Laplace (IPSL) Earth system model (ESM) and capable of simulating water, energy and carbon processes (Krinner et al., 2005). ORCHIDEE-MICT (aMeliorated Interactions between Carbon and Temperature) is a branch of ORCHIDEE with a better representation of high latitude process with new vertical soil parameterization, snow processes, and fires (Guimberteau et al., 2018). The recent ORCHIDEE-MICT v8.4.2 also includes modifications in wood harvest, forest age class and gross land use changes (Yue et al., 2018). The need to represent age-specific physiological and phenological characteristics for young and mature oil palm can thus benefit from this pre-existing forest age dynamics representation. Therefore, our development of oil palm modelling started from ORCHIDEE-MICT v8.4.2.

Processes related to the carbon cycle in ORCHIDEE include photosynthesis, respiration, carbon allocation, litterfall, plant phenology and decomposition (Krinner et al., 2005). We added a new PFT for oil palm starting from the default setting of the closest PFT —TBE trees. The major modification brought was for the carbon allocation, by including a new phytomer organ for oil palm, and a new fruit harvest module for fresh fruit bunch harvesting (Figure 2). The new model called ORCHIDEE-MICT-OP (oil palm) is schematized in Figure S2.

## 2.3 Introduction of phytomer structure

### 2.3.1 New phytomer structure

Oil palm has a monopodial architecture and sequential phenology. The phytomers are produced in succession, each bearing a big leaf with a number of leaflets, rachis and a bunch of fruits (Corley and Tinker, 2015; Fan et al., 2015). To represent the major morphology and phenological process, we introduce a new phytomer structure in the model frame. In the model, only branches and fruit bunches were specifically simulated at each phytomer while leaf was simulated as a whole of all phytomers at the PFT level to remain consistent with the four leaf age cohorts of the modelled phenological equations. Phytomers are initiated successively and developed in parallel on the same tree. Although each phytomer has its own sequence of initiation, allocation, fruit production and pruning, they share the same stem and root biomass and the same carbon assimilation process. In the default version of ORCHIDEE-MICT, there were eight biomass pools namely leaves, sapwood above and below ground, heartwood above and below ground, roots, seed and carbon reserve pools. To simplify the modification and parameterization of phytomers and keep consistent with the model structure, the branch and fruit bunch belonging to each phytomer were linked with the original sapwood and fruit biomass pools, although the fruit-bunch biomass pool was modified from the original model (Figure 2). The the number of fruit and branch component was set corresponding to phytomer number but the leaf linked with leaf biomass pool was divided to four age classes without duplication in each phytomers (Figure 2).

## 2.3.2 Phytomer phenology

Here we describe the phytomer dynamics related to planting, vegetative maturity and rotation at plant level and the sequential initiation and pruning at phytomer level. The modification of leaf seasonality is also presented. A schematic diagram of oil palm tree, phytomer and leaf phenology is shown in Figure 3. Since the phytomer phenology is closely related to the age of the tree, the age of the phytomer and the age of leaf, three temporal variables of tree age (the age of oil palm tree in years), the phytomer age (the age for each phytomer counted from its initiation, in days) and the leaf ages (the age of leaves in day) were used to compute tree, phytomer and leaf dynamics (Figure 3).

Based on the field evidence, there are three major phenological phases for phytomers during a tree life cycle. The first phase is the first two years between oil palm planting and the beginning of fruit-fill. In this period, leaf and branch begin to flourish and expand without fruit production. The second phase is the fruit development phase when fruit begins to grow and harvest begins, while fruit and branch biomass continue to increase. The third phase is the productive phase with high and stable yields that will last until the age of 25-30 years old. This phase ends up when the tree grows very tall (harvesting of fruit bunches becomes difficult) and the fruit yield starts to decrease. The modified subroutines of phytomer dynamics are adopted from the forest age cohorts simulated in ORCHIDEE-MICT v8.4.2. The forest age cohort module was originally designed for modelling forest management such as wood harvest and gross land use changes (Yue et al., 2018). This module allows us to represent photosynthesis, allocation and harvest practice for different forest age classes (each tree PFT is divided into 6 age 'cohort functional types' called CFTs) by setting CFT-specific parameters. This module is adopted to represent the rotation cycle of oil palm and the land conversion to or from oil palm. Here, the first phase of oil palm growth from age 0-2 is corresponding to CFT1, and the second phase corresponding to CFT2-4 starts from the end of age 2. The most productive phase is corresponding to CFT5 from age ~10-25 (Figure 3). Detailed parameterization for the new oil palm CFTs is presented in Section 2.4.

For an adult oil palm tree, the number of newly produced phytomers is stable at around 20-24 per year (Corley and Tinker, 2015). Phytomers are manually pruned twice a month to keep a maximum number of 40 phytomers, while fresh fruit bunches are harvested every 15-20 days (Combres et al., 2013; Corley and Tinker, 2015). Considering the regular development of phytomers and the periodic harvest and pruning practices, the initiation of new phytomers occurs every 16 days, and the phytomer longevity ($640=16\times40$, Figure 3) is set by this fixed initiation interval and by the maximum number of expanded phytomers of 40 in the model. Thereafter, we introduce two temporal variables in unit of days, i.e., the critical phytomer age or phytomer longevity ($Age_{phycrit}$) and the age of each phytomer ($Age_{phy}^{i,nphs}$). The former defines the time length between phytomer initiation and pruning, while the latter records the age of each phytomer. When the phytomer age reaches the critical value, the pruning practice is triggered and the pruned branch from phytomer and a group of old leaves from total leaf biomass go into the litter pool of the model. Subsequently, another new phytomer is initiated to maintain the total number of phytomers. The carbon allocation and harvest related to phytomer dynamics is discussed in the Sec. 2.3.3 and 2.3.4.

Leaf phenology of TBE forest is important for seasonal carbon and water fluxes. In another version of ORCHIDEE-MIC, the leaf phenology of TBE forests was implemented using four leaf age cohorts (See Figure 3) by Chen et al. (2020). Different photosynthetic efficiencies were used for leaf age cohorts to represent the leaf aging process. In this new canopy phenology scheme, NPP allocation to new leaves is driven by shortwave downwelling radiation ($SW_{down}$) and the vegetation optical depth of old leaves (Eq. 1 in Chen et al. (2020)), and weekly VPD is used to trigger the shedding of old leaves (Eq. 3 in Chen et al. (2020)). In the leaf shedding, the leaf longevity used in the VPD triggered leaf shedding scheme (eq. 2 and 3 in Chen et al., 2020) is modified to be the same than phytomer longevity (640 days) to approximate the old leaves removal in phytomers (it means than when all the 'leaves' dies, the phytomer dies). Here, we simplified the leaf growth without considering the "spear leaf" stage. We also ran a test simulation using a shorter $Age_{leafcrit}$ (620 days, Test1) in the supplement (Figure S8). The shedding leaf then enters to the litter pool. Here, we adopted this leaf phenology scheme for oil palm modelling.

### 2.3.3 Phytomer allocation

In ORCHIDEE-MICT, carbon is allocated to leaf, sapwood and root in response to water, light and nitrogen limitation (Krinner et al., 2005). The allocation of carbon to phytomers was simulated following this framework. The allocation to fruit and branch component for each phytomer was calculated as a fraction of the aboveground sapwood and the reproductive organ, whereas the allocation to leaves was unchanged. For each phytomer, the fraction of aboveground sapwood and reproductive organ allocated to branch and fruit components ($f_{br+fr}^{i,nphs}$, where *nphs* is the total number of phytomers and i is the index of phytomer) is a function of phytomer age as follows (Eq. 1). This fraction is further adjusted by the oil palm tree age to account for yield increase with tree growth ($F_{br+fr}^{i,nphs}$ Eq. 2).

$$f_{br+fr}^{i,nphs} = f_{br+fr,min} + (f_{br+fr,max} - f_{br+fr,min}) \times \left( \frac{Age_{phy}^{i,nphs}}{Age_{phycrit}} \times P_1 \right)^{P_2} \tag{1}$$

$$F_{br+fr}^{i,nphs} = f_{br+fr}^{i,nphs} \times \left(1 - \exp\left(-\frac{Age_{tree}}{P_3}\right)\right) \tag{2}$$

where $f_{br+fr,min}$ and $f_{br+fr,max}$ are prescribed values of minimum and maximum aboveground sapwood and reproductive organ allocation fractions to branch and fruit, which is increased with tree age. $Age_{phy}$ (day) is the age of phytomer, and $Age_{tree}$ (yr) is the age of the oil palm tree. $P_1$, $P_2$ and $P_3$ are empirical coefficients (set at 0.265, 2 and 0.8; unitless), respectively, based on yield calibration against observations). All abbreviations and parameter values are shown in Table S2. Note that the modifier ($f_{br+fr}^{i,nphs}$) range (0~0.07) is for one phytomer, and the total allocation fraction (a range of 0~1) should be the sum of modifiers in all phytomers.

After fruit initiation started (second phase, corresponded to CFT2-4), the allocation strategy changes with more resources shifted to fruit than leaf and the rate of fruit assimilation is accelerated (Corley and Tinker, 2015). This is represented by Eq. 1 with more carbon allocated to old and ripening phytomers to achieve the largest amount of yield. The further separation of

branch and fruit ($F_{br+fr}^{i,nphs}$) and fruit fractions ($f_{fruit}^{i,nphs}$) follows a similar scheme, i.e. an increase with phytomer age to accelerate fruit accumulation (Eq. 3).

$$f_{fr}^{i,nphs} = f_{fr,min} + (f_{fr,max} - f_{fr,min}) \times (1 - \exp(-Age_{phy}^{i,nphs} \times F_1)) \qquad (IF\ (Age_{phy}^{i,nphs} \geq ffblagday)) \qquad (3)$$

$$225 \quad f_{br}^{i,nphs} = F_{br+fr}^{i,nphs} - f_{fr}^{i,nphs} \qquad\qquad\qquad (4)$$

where $f_{fr,min}$ and $f_{fr,max}$ is tree age-specific value of minimum and maximum fruit allocation. $f_{br}^{i,nphs}$ stands for the branch fraction in the total branch and fruit fraction ($F_{br+fr}^{i,nphs}$), and $F_1$ is an empirical coefficient, set at 0.02 (unitless). The change of $f_{br+fr}^{i,nphs}$ and $f_{fr}^{i,nphs}$ with phytomer age is shown in Figure 3. The initiation of fruit begins when the phytomer age exceeds the pre-defined *ffblagday* (16 days). Also notice there is no fruit allocation during the first phase (CFT1).

The total phytomer allocation fraction is a sum of leaf, branch and fruit allocation:

$$f_{phy} = f_{leaf} + f_{sab+rep} \times \sum_{nphs}^{i} F_{br+fr}^{i,nphs} \qquad\qquad (5)$$

where $f_{leaf}$ is the leaf fraction, and $f_{sab+rep}$ is the aboveground sapwood and the reproductive organ allocation fraction, respectively.

### 2.3.4 Fruit harvest

The default wood harvest in ORCHIDEE-MICT is based on the different forest age classes (implemented as CFTs). For each CFT, when the stem biomass reaches the prescribed maximum woody biomass of current CFT, it will move to the next CFT. Wood harvest can start from any CFT by user's choice, and the default wood harvest sequence starts from the second youngest CFT to the oldest one and back to the youngest until reaching the required harvest amount (Yue et al., 2018). Unlike wood harvest, oil palm fruit is produced in sequence and harvested regularly. Here we assume the harvested fruits were taken from 240 the oldest phytomer before pruning. The duration between fruit initiation and harvest is prescribed ($Age_{ffbcrit}$ (day), Table S2), and fruits will be harvested after the phytomer age in the oldest phytomer reaches the $Age_{ffbcrit}$. The harvested fruit biomass is then added to a new separate harvest pool.

### 2.4 Parameter calibrations for oil palm

Since most parameters vary across different PFT, we systematically adjusted parameters related to photosynthesis, respiration, 245 carbon allocation and morphology for oil palm according to the observed values from field measurement literature. Some parameters are CFT-specific values in accordance with the tree age cohorts in the model. Details of the parameters for oil palm are summarized in Table S2.

### 2.4.1 Photosynthesis parameters

The photosynthesis module of ORCHIDEE-MICT is based on an extended version (Yin and Struik, 2009) of the Farquhar, von Caemmerer, and Berry model (FvCB model; Farquhar et al., 1980). Leaf age class is introduced to take into account the fact that the photosynthetic capacity depends on leaf age (Ishida et al., 1999). The maximum rate of Rubisco activity ($V_{cmax}$) is defined by the prescribed $V_{cmax25}$ and weighted leaf efficiency ($e_{rel}$, unitless: 0–1). The relative leaf efficiency ($e_{rel}$) is a function of relative leaf age ($A_{rel}$) where $A_{rel}$ is the ratio of the leaf age to the critical leaf age (the same as $Age_{phycrit}$), also known as leaf longevity (Figure 4, red line). The $e_{rel}$ change with $A_{rel}$ in the default ORCHIDEE-MICT version is shown in Figure 4 (black dashed line), which increases from a low initial value to 1 (reaching the prescribed optimal $V_{cmax25}$) for a given period and then decreased to a low level for the old leaves. This was modified by setting the minimum efficiency to 0 and at both leaf flushing and longevity based on observations of the leaf phenology of Amazonian TBE forest in another ORCHIDEE-MICT version with leaf cohorts, ORCHIDEE-MICT-AP (blue dashed line) (Chen et al., 2020). However, unlike the natural TBE forest, the old leaves in the old phytomers of oil palm are probably more productive to sustain the high fruit amount because of the sequent growth, phytomer pruning and fruit harvest. Thus, $e_{rel}$ for the old leaves of oil palms is maintained the same as the value in the default ORCHIDEE-MICT version (red line in Figure 4). We also adjusted $V_{cmax25}$ for each tree age class of oil palm according to the experimental evidence (Fan et al., 2015; Meijide et al., 2017; Teh Boon Sung and See Siang, 2018) (Table S2). $V_{cmax25}$ for oil palm increases with tree age (from 35 to 70 μmol m$^{-2}$s$^{-1}$) corresponding to the increase of gross assimilation (Breure, 1988). Another two important parameters for photosynthesis are maximum leaf area index (LAI$_{max}$, controlling the maximum carbon allocation to leaf biomass) and specific leaf area (SLA). The observed maximum LAI varies from 4 to 7 m$^2$m$^{-2}$ across different genotypes, plant densities and soil types (e.g., peat) according to nine observation-based publications listed in Table S2, and LAI$_{max}$ was found to increase with oil palm tree age (Kallarackal, 1996; Kotowska et al., 2015; Legros et al., 2009). SLA, by contrast, generally decreases with oil palm tree age from 0.0015 to 0.0008 m$^2$g$^{-1}$C (Van Kraalingen et al., 1989; Legros et al., 2009; Kotowska et al., 2015). We thus used a CFT-specific value which is close to the median values of LAI and SLA obtained from observational data (Table S2).

### 2.4.2 Respiration parameters

Autotrophic respiration (AR, including maintenance and growth respiration, MR and GR) in ORCHIDEE-MICT is based on the work of Ruimy et al. (1996). MR is a function of the temperature and biomass for each plant part (Eq. 6-7) whereas GR is prescribed as 28% of the allocable assimilates for TBE tree PFT (Krinner et al., 2005). Field evidence shows that MR in gross assimilation of palm increases with oil palm tree age but MR per unit of tree biomass decreases (Breure, 1988). In total, AR represents 60-75% of GPP for oil palms (Henson and Harun, 2005). Based on this prior knowledge, we adjusted both the constant $S_1$ in Eq. 7 and the fraction of GR in GPP ($f_{GR}$). The former parameter ($S_1$) increases with age and the latter does the opposite ($f_{GR}$) (Table S2). The parameter values were calibrated to match the observation of GR/MR, AR/GPP and GPP.

$$MR_j = Biomass_j \times C_{0,j} \times (1 + slope \times T) \tag{6}$$

$$slope = S_1 + S_2 \times T_l + S_3 \times T_l^2 \qquad (7)$$

Where $j$ is the different plant part. $C_0$ is prescribed for each plant part for each PFT. $T$ is the 2-m temperature/root temperature for above/belowground compartments. $T_l$ is the long-term (annual) mean temperature. $slope$ is the second-degree polynomial dependency of $T_l$. $S_1$, $S_2$ and $S_3$ are empirical coefficients.

### 2.4.3 Carbon allocation parameters

Carbon allocation to new leaves in the ORCHIDEE-MICT-OP was modified following the ORCHIDEE-MICT-AP by Chen et al. (2020) as described in 2.3.2. The leaf allocation ($f_{leaf}$) is both related to the amount of sunlight available at the top of canopy and the light transmission of old leaves so that the $f_{leaf}$ is expressed as a function of higher shortwave downwelling radiation ($SW_{down}$) and LAI of the old leaves as followings:

$$f_{leaf} = f_{leaf,min} + (f_{leaf,max} - f_{leaf,min}) \times (SW_{down} \times e^{-L_1 LAI_4}/L_2)^{L_3} \qquad (8)$$

where $f_{leaf,min}$ and $f_{leaf,max}$ are the prescribed values for minimum and maximum leaf allocation. $LAI_4$ is the LAI of the oldest leaf age cohort 4. $L_1$, $L_2$ and $L_3$ are empirical coefficients, setting to be 0.45, 100 and 3 (unitless), based on the calibrations using observed NPP allocation among leaf, sapwood and fruit (Henson and Dolmat, 2003; Van Kraalingen et al., 1989).

The original leaf ($f_{leaf,ori}$), root ($f_{root,ori}$), and sapwood and reproductive tissue ($f_{sap+rep,ori}$) allocation scheme in response to the water, light and nitrogen in the ORCHIDEE-MICT-OP was modified from the default ORCHIDEE-MICT. To harmonize the new leaf allocation fraction ($f_{leaf}$) and the original one ($f_{leaf,ori}$), root, sapwood and reproductive organ allocation fractions were further rescaled:

$$f_{root} = max[min[f_{root,ori} - R_1 \times abs(f_{leaf} - f_{leaf,ori}), f_{root,max}], f_{root,min}] \qquad (9)$$

$$f_{sap+rep} = 1 - f_{root} - f_{leaf} \qquad (10)$$

Where $f_{root,min}$ and $f_{root,max}$ is the prescribed values of minimum and maximum root allocation according to Kotowska et al. (2015). $R_1$ is an empirical coefficient (= 0.95).

NPP partitioning between aboveground part of sapwood, reproductive organ and belowground sapwood biomass is a function of tree age. Older trees get more allocation to aboveground part than younger ones (Krinner et al., 2005). In the default ORCHIDEE-MICT version, the values of minimum and maximum NPP partitioning to aboveground biomass are constant. By contrast, observed oil palm gross assimilation increases with age (Breure, 1988), and most of the assimilates go into phytomer to sustain fruit production. In ORCHIDEE-MICT-OP, we adopted the original model equation of allocation to aboveground sapwood and reproductive organ ($f_{sab+rep}$) increasing with age (Eq. 9) but adjusted parameters to match the observations.

$$f_{sab+rep} = f_{sab+rep,min} + (f_{sab+rep,max} - f_{sab+rep,min}) \times (1 - e^{\frac{-Age_{tree}}{\theta}}) \tag{11}$$

Where $f_{sab+rep,min}$ and $f_{sab+rep,max}$ are prescribed tree age-specific values of minimum and maximum allocation to the aboveground sapwood and the reproductive organ, which increases with tree age. $Age_{tree}$ is the oil palm tree age, and $\theta$ is the empirical CFT-dependent coefficients (Table S2).

### 2.4.4 Other parameters

Other adjustments of parameter values include morphological, phenological and turnover parameters. The maximum number of phytomer (*nphs*) is set as 40 according to observations (Combres et al., 2013; Corley and Tinker, 2015). Given the phytomer initiation rate of 20-24 per year, the pruning frequency of twice a month and the number of phytomer (Combres et al., 2013; Corley and Tinker, 2015), the critical phytomer age ($Age_{phycrit}$) is estimated to be around 600 to 720 days. Based on previous studies (Van Kraalingen et al., 1989; Corley and Tinker, 2015; Fan et al., 2015), the leaf longevity for oil palm is 600-700 days, shorter than the 730 days used for the default TBE tree PFT in ORCHIDEE-MICT. As a result, both the critical leaf age (leaf longevity) and the critical phytomer age ($Age_{phycrit}$) are set to be 640 days. The critical fruit age ($Age_{ffbcrit}$), defined as the duration between the fruit initiation and harvest, is set as 600 days, that is, shorter than the critical phytomer age, allowing leaf senescence after fruit harvest.

After pruning, cut branches in a pruned phytomer are transferred to the litter pool. Considering that the removal of leaves is not very well represented at the time of phytomer pruning, we further added an extra leaf loss ($Loss_{leaf}^m$) of the old leaves (using the leaf age cohort) at the time when the oldest phytomer is manually pruned as follows:

$$Loss_{leaf}^m = Biomass_{leaf}^m \times LO_1/nphs \text{ (m=3,4)} \tag{10}$$

Where $Biomass_{leaf}^m$ is the leaf biomass for leaf cohort *m*, $LO_1$ is an empirical leaf loss coefficient. A test with VPD-triggered leaf shedding excluding the extra leaf loss during the phytomer pruning was also performed in supplement as comparison.

In the default ORCHIDEE-MICT version, carbon residence time ($\tau$) of biomass is set as 70 years for natural tropical forests to represent the natural mortality. Oil palms, on the other hand, are managed are clear-cut at ~ 25 years for the next rotation cycle. The natural tree mortality is thus not applicable for oil palms. In ORCHIDEE-MICT-OP, we assumed that oil palm is manually cut down for rotation before the natural mortality without considering the disease and other causes of tree loss as well (clear-cutting every 25 years, Figure 5).

### 2.4.5 Sensitivity analysis

Because of the distinct age cohorts of oil palm and age-based parameterizations for photosynthesis and allocation in ORCHIDEE-MICT-OP, performing the sensitivity analysis on every age-specific parameter would be too CPU intensive. Instead, we performed sensitivity tests of the major parameters related to oil palm photosynthesis and allocation, particularly for the phytomer related allocation parameters without enough constraints from field observations. For the age-specific

parameters (e.g., $V_{cmax25}$, $sla$), the calibrated value for CFT5 (the most productive phase with the maximum yield) were tested. The sensitivity tests were conducted by changing the selected parameters (variables with * in Table S2) by ±5, ±10 and ±20%

from the originally calibrated value while keeping the other parameters unchanged. Their impacts on the cumulative yields at the most productive phase aging from 10-25 (corresponded to CFT5) were evaluated. For the grouped parameters such as the phytomer allocation coefficient ($P_1/P_2/P_3$), the sensitivity was tested by changing ±5, ±10 and ±20% of the target function ($F_{br+fr}^{i,nphs}$) using different combinations of $P_1 \sim P_3$.

**2.5 Site simulation setup**

The 6-hourly 0.5° global climatic data, CRUNCEP v8 and the 0.08° global soil texture map were used as forcing data in the simulations (Reynolds et al., 2000). The vegetation cover of the 14 sites (Figure 1 and table S1) was all set to the oil palm PFT with a coverage of 100%. Biomass boundary value for each age classes (Figure 5) are prescribed for oil palm based on the prior knowledge from observation (Tan et al., 2014). When the total biomass reaches the lower boundary of the oldest tree age class (CFT6, Figure 3 and 5) and moves to CFT6, wood harvesting will be performed, and oil palm trees will thus be cut down.

New oil palms will be established in the youngest tree class (CFT1) for the next rotation cycle. Site simulations were run for 30 years which is consistent with the rotation duration of ~25 yrs and the climatic forcing for the period between 1986 to 2015 were used. Spin-up simulation was not performed since we didn't focus on the soil organic carbon and there is no feedback of soil carbon to plant growth in the model. Oil palm yields at maturity were calculated using the average values during 11-20 years for comparison. Fruit yields are converted to kg DM ha$^{-1}$yr$^{-1}$ using a carbon ratio of 0.45.

**3 Results: model evaluation**

**3.1 LAI and Leaf phenology**

Figure 6 shows annual dynamics of observed and simulated LAI vs. tree age averaged over the 14 observation sites (black line). For each age, we collected observational LAI values from different field measurement studies and presented the medians and ranges (the red marker and error bar) in Figure 6. Since there are no continuous LAI measurements available (to the best

of our knowledge), we combined single LAI measurements at a certain age from different studies. The simulated LAI increases from 0.3 to 5.3 in the first ~10 years, and then stays stable at the maximum value (5.5, Figure 6). The simulated LAI trajectory can generally reproduce the trend from observations. Although simulated LAI ranges overlap with the ranges of LAI observations at most ages, some observations are not reproduced at Age 13 and Age 19 when the model achieved a stable and maximum LAI (Figure 6). This variability of LAI measurements reflects the use of different sites with different oil palm

species and management practices. In the model, however, genotypes and practices are uniform. The detailed intra-annual variations of LAI, combined with leaf biomass and $V_{cmax}$ for each leaf age cohort are shown in Figure S3 with significant seasonality after merging the leaf phenology scheme from Chen et al. (2020). Compared to the ORCHIDEE-MICT version

with no seasonality in LAI (dashed line in Figure S3a), the LAI of young leaves increases but decreases for old leaves during the canopy rejuvenation period (January to May, solid line in Figure S3a). The opposite behavior is shown in the rest of the year. Similarly, the default ORCHIDEE-MICT version shows no seasonality of leaf age and leaf photosynthetic efficiency in different leaf age classes (dashed line in Figure S3b and c), while the seasonality of leaf age and leaf efficiency is successfully captured in this version (solid lines in Figures S3b and c).

## 3.2 Productivity and fruit yield

The simulated GPP, NPP and fruit yield in comparison with field measurements are shown in Figure 7. Compared to the default ORCHIDEE-MICT version, NPP can be better reproduced by ORCHIDEE-MICT-OP (solid squares closer to 1:1 line than open square, Figure 7a) with a Normalized Mean Bias Error (NMBE, defined as the sum of biases divided by the sum of field values) of 12.87% and $r^2$ of 0.9 across sites. Among the 14 sites with NPP observations, simulated NPP at Site 1, 7 and 12 is comparable with observations with a NMBE of only 4.0% while simulated results from other sites are relatively higher than observations (NMBE of 28.8%). For GPP, there are only three observations available, and simulated values by ORCHIDEE-MICT-OP are relatively higher than the observed values with a NMBE of 25.4%.

For fruit yields, we collected six single-year observations at different sites for oil palm plantations aged from 10-15 yrs, expect for one site where yield data cover ages 4 to 16. The observed oil palm yields at maturity vary from 13.0 to 22.1 t DM ha$^{-1}$ yr$^{-1}$ with a median of 15.0 t DM ha$^{-1}$ yr$^{-1}$, and the simulated yields show a similar range of 12.2-21.4 t DM ha$^{-1}$ yr$^{-1}$ with a median of 16.9 t DM ha$^{-1}$ yr$^{-1}$. Thus, simulated fruit yields show an overall good agreement with site observations with a NMBE of 6.1% (Figure 7c). There is only one site (Site 3) with available yield estimates for successive years (Figure 7d). It should be noted that it is not real observations but a fitted curve with oil palm age of yield data provided by the Malaysian Palm Oil Board (MPOB) research station at Keratong (Tan et al., 2014). This yield-age curve shows a strong yield increase after Age 10 and even Age 25 (Figure 7d), which is against the field evidence that fruit yields for oil palms reach maximum at ~10 yr, stay relatively stable, and decrease after ~25 yr (Boo et al., 1994; van Ittersum et al., 2013). The reduction in yields after ~25 yr is also one of the reasons for clear cutting for next rotation. Still, we compared our simulated yields with that yield-age curve (Figure 7d). Simulated annual fruit yield at Site 3 is generally consistent with data during the first 9 years but lower than the curve in the subsequent years, probably due to the uncertainties in the yield-age curve. Besides, the simulated annual and cumulative yields also showed good agreement with observations in the two independent sites (site in Merlimau estate in Figure 11, Teh and Cheah 2018 and site PTPN-VI in the Figure 6, Fan et al., 2015), indicating the model's ability to capture yield dynamics (Figure S6 and S7).

## 3.3 Biomass

Figure 8 shows the comparison of simulated biomass and time series with observations. The biomass here includes the developing fruit but exclude the harvested fruit biomass. Note that some sites have several observed values (Site 1, 2, 9 and

10 in Figure 8a) at different age and for biomass components e.g., total biomass (TB), above ground biomass (AGB) and below
ground biomass (BGB). A total of 13 biomass observations were collected at different age groups (3 in the young age group, 8 at maturity and the remained 2 for averaged biomass among several years, Table S1). Compared to the default ORCHIDEE-MICT version, simulated biomass by ORCHIDEE-MICT-OP is more consistent with observations (Figure 8a). 10 out of 13 sites, are distributed close to the 1:1 line except Site 2 (TB at age 10), Site 9 (AGB at age 16) and Site 10 (AGB at maturity). The NMBE of oil palm biomass is 10.4% after excluding Site 9 with the largest bias, compared with the 156.7% by the default ORCHIDEE-MICT. We further compared the simulated above, below ground biomass and their ratio with observations (Figure 8b). Similarly, the ORCHIDEE-MICT-OP version can better reproduce the observations than the default ORCHIDEE-MICT version. The NMBE for above and below ground biomass between ORCHIDEE-MICT-OP and observations are 12.1% and 55.3%, respectively. The ratio of AGB and BGB is calculated at 1.7, which is much closer than the observation (1.1-3.0) compared with that of default ORCHIDEE-MICT (0.7-0.8).

There are only two sites (Site 3 and 12, Figure 8c and d) with time series of biomass. Similar to the fruit yields (Figure 8d) simulated biomass by ORCHIDEE-MICT-OP generally agrees with observed values but is higher in the first 18 years and lower afterward (Figure 8c). At Site 12, ORCHIDEE-MICT-OP simulated biomass is higher than observations for the whole oil palm life cycle. This is probability because Site 12 was covered by very deep peat soil (>3m) with a high soil water table and high C density and the potential impact on the oil palm production is not considered (e.g., different nutrient availability in peat and mineral soil and palm leaning in peat soil which may cause the decline of yield). A detailed discussion of the oil palm on peat is presented in Section 4.2 and 4.3. Also, the calibration is based on the observations from all sites and no calibration was applied for this site, which may cause the higher estimation. The NMBE is 16.2 % and 15.5% at Site 3 and Site 12. The default ORCHIDEE-MICT version largely overestimated the biomass at both sites (dashed line in Figure 8c,d).

### 3.4 Partitioning of GPP, NPP and Biomass

Comparison of oil palm GPP and biomass partitioning between simulations and observations is shown in Figure 9. Compared to the default ORCHIDEE-MICT version (grey bars), simulated results from the ORCHIDEE-MICT-OP version (black bars) are closer to the observations (red bars, Figure 9). GPP is partitioned to GR, MR and NPP whereas NPP is further divided into allocation to stem and frond, root and fruit (Figure 9a). The simulated growth and MR fraction in GPP ranges from 17.1-28.8% and 28.1-54.3% respectively, which is comparable with observations (21-31% and 34-44%) from Henson and Dolmat (2003). The simulated fraction of autotrophic respiration in GPP (60.87%) is also consistent with the observed fraction (60-75% (Henson and Harun, 2005)). In the simulation by ORCHIDEE-MICT-OP, stem and leaf (median of 18.9% in GPP) occupies the largest parts of NPP, followed by fruit allocation (17.5%) and root allocation (2.8%). The differences between the simulated NPP fraction for stem and leaf, root and yield by ORCHIDEE-MICT-OP and observed fraction are 10.9%, -1.4% and -2.0%, respectively, indicating a good representation of NPP allocation to different biomass components in the new model.

Simulated partitioning of biomass by ORCHIDEE-MICT-OP is closer to observations (Breure, 1988; Henson and Dolmat, 2003; Tan et al., 2014) than the default ORCHIDEE-MICT version (Figure 9b). The simulated leaf and root and other organs (stem, fruit and branch biomass) proportion of total biomass varies between 51.7-75.1%, 14.7-32.4%, and 8.5-16.0%. The simulated fraction to other organs is higher (14.7%) than observations, and correspondingly it is lower for leaf (-6.1%) and root (-5.6%) fractions, the improvements reaches 18.8%, 13.0% and 6.2% compared to the biases in the default ORCHIDEE-MICT. Note that the proportion of fruit bunch and branch of a phytomer is not separated but added in the stem proportion because most of the studies presented fruit and branch biomass fraction as a part of stem biomass (Van Kraalingen et al., 1989; Henson and Dolmat, 2003). Also, the time and frequency of collecting fruits and measuring biomass are usually not synchronous. There is only one field study showing that the phytomer (fruit and branch) fraction varies between 5.0-14.5% of the total biomass after fruit harvest (Breure, 1988), which is comparable with the simulated median proportion of 14.4% by ORCHIDEE-MICT-OP.

### 3.5 Phytomer development

Growth of phytomers during the life cycle (initiation, fruit development and productive phases) of oil palm is presented in Figure 10. Figure 10a and b show the fruit and branch growth in single phytomer (8 in 40 phytomers were shown for a better visualization), while figure 10c is the total biomass for all the 40 phytomers as a sum of leaf, branch and fruit components. The initiation phase roughly corresponds to the oil palm tree age between 0 to 2 without any fruit production. Subsequently, age 2-10 is the fruit development phase. After 10 years old, oil palm reaches the productive phase with the maximum and steady fruit yields. This phenological characteristic is consistent with the oil palm development observed in previous studies (Sunaryathy et al., 2015). Some study even shows the productive phase can start as early as ~7 year old (Henson and Dolmat, 2003).

Biomass of leaf and branch of all the phytomers starts to increase after planting (Figure 10c) and reaches about 211.3 and 28.6 gC m$^{-2}$ at the end of age 2. The fruit production and harvest begin after entering the fruit development phase (the end of age 2) (Figure 10a), whereas the total fruit biomass increases rapidly to 367.6 gC m$^{-2}$ yr$^{-1}$ at age $\approx$ 10. From age 2 to 10, phytomer biomass increases with a stair-step shape, and fruit and branch biomass slightly decline when moving from one tree age class to the next older class. This is because values for some parameters (e.g., V$_{cmax}$ and LAI$_{max}$, Table S2) are different among the CFT 2-4 in the fruit development phase. For example, LAI$_{max}$ increases from 3.5 in CFT3 to 4.5 in CFT4. In the ORCHIDEE framework, biomass will preferentially allocate to leaf to reach LAI$_{max}$ in order to grow more leaves to increase GPP and then allocate to other biomass parts when LAI reaches LAI$_{max}$ (Krinner et al., 2005). Therefore, when oil palms move from CFT3 to CFT4, the increased LAI$_{max}$ drives more biomass going to leaf (Figure 10c) and less to fruit and branch at the beginning of CFT4, resulting in the small decline in the fruit and branch biomass. We acknowledge that this model behavior may contradict the reality, but the small magnitude and short duration of declining (Figure 10c) may have little impact on the modeling results. At the productive (maturity) phase after age 10, the average leaf, fruit and branch biomass are 683.8, 424.0 and 64.8 gC m$^{-2}$, which consists of 58.3%, 36.1% and 5.5% of the total phytomer biomass (40 in total), respectively.

### 3.6 Sensitivity analysis results

The maximum rate of carboxylation ($V_{cmax25}$) is the most sensitive photosynthesis parameter because it determined the photosynthesis rates of leaf, followed by *sla*. Changes in ±20% of the baseline value of $V_{cmax25}$ leads to 13.8%/20.5% increase/decrease in the cumulative yields from age 10 to 25 (Figure 11). Maximum leaf area index ($LAI_{max}$), a threshold beyond which there is no allocation of biomass to leaves, has a smaller influence on the yields than $V_{cmax25}$ and *sla*. Yields are not changed linearly with changes in the $LAI_{max}$ value since it is a threshold parameter by definition.

For the allocation parameters, the empirical coefficients for the leaf ($L_1$/$L_2$/$L_3$) (Eq. 8) and root ($R_1$) (Eq. 9) allocation have very small impact on the fruit yields. The other allocation parameters are more or less related to the NPP allocation to aboveground sapwood and the reproductive pool, which influence the dynamics of the phytomer biomass and fruit yields. Among these parameters, yields are most sensitive to the phytomer allocation coefficients ($P_1$/$P_2$/$P_3$) (Eq. 1 and 2) which determine the NPP partitioning to phytomer (10% decrease in ($P_1$/$P_2$/$P_3$) leads to a decline of 21.23% in yield). The $f_{sab+rep,max}$ parameter controls the upper boundary of allocation to the aboveground sapwood and the reproductive organ (Eq. 11) and brings 19.4% increase in yields by changing +20% of the default value. Similarly, increasing/decreasing (10%) maximum fresh fruit bunch allocation fraction ($f_{fr,max}$) results in a significant increase/decrease (10%) of yields. By contrast, changing the baseline values of $f_{sab+rep,min}$ , $f_{fr,min}$, $F_1$ (fruit bunch allocation coefficient), $\theta$ (the coefficient of partitioning allocation between above and belowground sapwood) and *ffblagday* leads to little influence on the final cumulative yields. The turnover-related parameter $LO_1$ exerts a negative impact on cumulative yields. The increase of $LO_1$ increased the old leaf loss throughout phytomer pruning and results in lower yield.

## 4 Discussion

### 4.1 Model performance before and after oil palm implementation

Based on the default ORCHIDEE-MICT version and the leaf age cohort scheme in the ORCHIDEE-MICT-AP version, the oil palm PFT has a new phytomer organ and a yield harvest pool (Figure S2), with other model parameters recalibrated. The new ORCHIDEE-MICT-OP version allows for simulating oil palm morphology, phenology, biomass growth and yields. We evaluated the LAI, GPP, NPP, yields and biomass of oil palm in ORCHIDEE-MICT-OP using available observations from previous field measurement studies (Table S1).

In the default ORCHIDEE-MICT version, oil palm is taken as TBE tree PFT, which causes biases in the simulation. For example, it is impossible to realize regular fruit harvest and phytomer dynamics in the default ORCHIDEE-MICT version without the phytomer structure and the fruit harvest pool. The introduction of phytomer structure and the sequential developing processes allows for reproducing variable developmental stages for each phytomer including the initiation, fruit production, harvest and pruning in the model. Besides, the modification of carbon allocation scheme improves allocation of the assimilated carbon and partitioning of biomass pools (Figure 9). Oil palm trees have specific physiological characteristics which are

different from other tropical forests. The evolution of physiology with age is implemented by new tree age-specific parameterization scheme based on the tree age cohort module of ORCHIDEE MICT. Carbon assimilation is accelerated with increasing oil palm age. Carbon allocation to phytomer shifts more resources to fruit than leaf and branches as fruits mature. Consistent with observations, the fruit yields also show an increase from young to old trees. To our best knowledge, distinct age classes of oil palm and the age-based parameterizations for photosynthesis and autotrophic respiration dynamics have not yet been implemented in the previous LSMs aiming to simulate oil palm biophysical variables. The leaf age cohort-based phenology scheme from ORCHIDEE-MICT-AP was also adapted for oil palms to improve the seasonality of leaf and photosynthesis (Figure S3). This process was not included in any previous oil palm models either. Moreover, the calibration for age-specific parameters is based on the 14 individual observation sites with variable climate and soil conditions and we also compared the simulation results with observations for a range of variables including biomass, yield, LAI, GPP and NPP and biomass/GPP component. Therefore, our parameterizations of oil palm (Table S2) can also be a reference as for other LSMs.

## 4.2 Uncertainty in the model

Although the simulation of oil palm shows a significant improvement in the new model, there are some limitations in this version. The growth of oil palm is simplified to be incorporated into the model structure. For example, we assumed a constant maximum phytomer number of 40 for each oil palm through its whole life cycle. However, the expanded phytomer number may decrease with age according to some studies, and the maximum number is lower than the actual value in some areas (e.g., 32) (Corley and Tinker, 2015). The maximum number of phytomers is externalized as an input parameter in the model, making it flexible to be changed by users' choice. Some factors related to oil palm yields such as the gender of inflorescence and the rate of inflorescence abortion are not considered because of the limited understanding of underlying mechanisms (Breure and Menendez, 1990; Henson and Mohd, 2004). Instead, a simplified structure of one phytomer carrying one fruit bunch is used. Also, considering the oil palm is a highly manged plantation unlike natural forest, some rigid parameterization is adopted such as phytomer initiation interval, fruit harvest interval, phytomer pruning interval and leaf longevity. According to the field observations, the average temperature of the coldest month of the year for oil palm growth should not fall below 15 °C, and the optimal temperature condition ranges between 24 and 28 °C (Corley and Tinker, 2015). Oil palm stomata began to close when air temperature rose above 32°C (Rees 1961). In the main oil palm growing areas, temperatures are relatively uniform throughout the year (fluctuated at ~27°C) and rarely falling below 22°C (see the monthly temperature variations in Figure S9). Therefore, growing degree day and low temperature may not be the major limitations for oil palm growth. In addition, regular harvest and pruning practice (about twice a month) is conducted in the commercial oil palm plantations, which regulates the total number of phytomers. Based on these, the phytomer initiation in sequence is determined by a fixed time interval (16 days). This assumption in our model is thus a balance between the plant growth and human management practices. A previous study also used the period of thermal time (Fan et al., 2015) to regulate the phytomer initiation.

The accessibility and data sources of observations also vary from site to site, which influence the calibration of parameters and the evaluation of model performance. Without direct annual observations for parameters related to LAI and autotrophic respiration, some age-specific parameters are empirically calibrated based on multiple observations like GPP, NPP and biomass. The observations used for calibration and evaluation such as yields, biomass and GPP also vary from genotypes,

management practices, and measurement methods. For example, the annual fruit yield data in Site 3 (red line in Figure 7d) is a fitted curve using fruit yields from a nearby research station (Tan et al., 2014) while some others are measured fruit weight after fruit harvest every time (Henson 2003). Destructive and non-destructive based methods were used to obtain the AGB for different sites, and different allometric equations applied in the non-destructive based method may cause up to 10% biases (Corley and Tinker, 2015). In site 6, the simulated GPP by ORCHIDEE-MICT-OP is 50% higher than the observed value. The

mismatch between model and observation may also be caused by the uncertainty in observations or non-resolved soil fertility effects. Specifically, since the model can generally capture NPP (simulated NPP = 1700 gC m$^{-2}$ yr$^{-1}$ at Site 1; only  Site 1 has both GPP and NPP observations), and the proportion of autotrophic respiration in GPP is 60-75% (Henson and Harun, 2005), the estimated GPP at site 1 should be 4256.6-6810.5 gC m$^{-2}$ yr$^{-1}$, much higher than the observed value of ≈ 3360 gC m$^{-2}$ yr$^{-1}$. Moreover, yield of oil palm usually ranges from 587 to 996 gC m$^{-2}$ yr$^{-1}$, so the low observed GPP at site 1 may not be consistent

with this yield range. Factors such as genotypes, management practices (excepted fruit harvest and phytomer pruing) and plantation scales that influence oil palm biomass and fruit yield are not fully included in the model, and thus it is impossible to perfectly reproduce the all site-level observations using our model. The reported fruit yields of different genotypes vary from 114.4-112.2 kg plant$^{-1}$ yr$^{-1}$ to 81.7-98.5 kg plant$^{-1}$ yr$^{-1}$ in Kandista and Batu Mulia (Lewis et al., 2020), and leading plantation companies in Indonesia and Malaysia have achieved average fruit yields of 173.7 kg plant$^{-1}$ yr$^{-1}$ (Donough et al.,

2009). The amount and types of fertilizers used in oil palm plantation also vary from site to site. In some area, applied fertilizer amount is according to the leaflet nutrient contents while regular fertilization was applied in some other places (Legros et al., 2009; Kotowska et al., 2015). In the current ORCHIDEE-MICT version, however, nitrogen and phosphorus cycles are not explicitly included, limiting the implementation of fertilization effects on plant growth in the model. The scales of plantation also impact oil palm biomass and yields due to the differences in managements (e.g. dedicated managements in the large

industrial plantation and extensive practices in smallholders). Another important factor is the difference between oil palms grown on mineral and peat soils. Although our model generally was able to reproduce the yield, GPP and NPP at one peat-based oil palm site (Site 12), the biomass is overestimated throughout the life cycle, indicating further work is needed to implement the peat oil palm in the LSMs (and other data from peat soils for yields). Previous studies suggested that the frond biomass of oil palm grown on peat soils was lower than on mineral soils in all age classes (Henson 2005). On peat soil, oil

palm allocates less biomass to root system (Corley, Gray and Kee 1971; Othman et al., 2010). Further decomposition of peat subsidence after peatland drainage combined with poor anchorage of oil palm may cause palm leaning and even palm falling and hence increase mortality (Henson et al., 2003; Othman et al., 2010). Based on the yield and tree mortality, the rotation cycle also varies in mineral- (25-30 years) and peat- (18-20 years) based oil palm. A better representation of peat oil palm could be reached by using a separate parameterization scheme for peat oil palm (e.g., adjusting the partition between AGB and

BGB and decrease the carbon assimilation rate), adopting a lower biomass threshold for oil palm rotation (Figure 5), modifying the carbon emission rate at the beginning years of oil palm conversion and so on. However, it would be a great challenge to implement some factors such as disease in the current stage without enough knowledge on the processes and impacts of disease on oil palm growth. Also, we note the optimal planting density is different between the two soil types (110-148 palms ha$^{-1}$ on mineral soil and 160-200 palms ha$^{-1}$ on peat soil) (Henson et al., 2003; Othman et al., 2010; Lewis et al., 2020). The mineral-

based oil palm suffers a decline in frond biomass and production while that of the peat oil palm is less influenced (Lewis et al., 2020). These would also cause biases in simulated biomass and yield due to no separation between mineral- and peat-based oil palm.

## 4.3 Implication and application of ORCHIDEE-MICT-OP

The newly developed ORCHIDEE-MICT-OP can be a useful tool to predict future oil palm yields, simulate LUC carbon

emissions and estimate impact on ecosystem services. Malaysia and Indonesia experienced the highest oil palm expansion (3.8 and 9.7 million ha) over the world from 2001 to 2016 (Xu et al., 2020).The drainage and replacement of peatland (3.1×10$^6$ ha, 27%) in Malaysia and Indonesia by oil palm expansion turned this carbon-rich region to a carbon source (Miettinen et al., 2016). It is thus important to simulate the carbon budget and calculate the carbon changes after oil palm expansion. Previous studies calculated the potential carbon emissions from forest conversion by oil palm using a uniform carbon density value

without considering spatial heterogeneity and temporal variations (Carlson et al., 2013; Cooper et al., 2020). In reality, the biomass loss from deforestation is fast but soil carbon change may take a long time in mineral soil. A more complex condition would happen in the conversion to oil palm plantation on the peat soil, where huge carbon emission was observed in the first 5 years following conversion (Hooijer et al., 2012; Cooper et al., 2020). Based on the framework of gross land use changes, the grid-based ORCHIDEE-MICT-OP could thus contribute to the quantification of spatial and temporal dynamics of LUC

carbon emissions from oil palm expansion. Moreover, one of the ORCHIDEE branches, ORCHIDEE-PEAT, has already implemented the peat processes for high latitudes (Qiu et al., 2018). Merging the oil palm specific morphology, phenology and harvest processes of oil palm and the peat related processes in these two branches would help characterize the oil palm yields as well as carbon, water and energy fluxes on peat soil palms. Given the high rate of oil palm expansion in Malaysia and Indonesia, there is an urgent need to evaluate the potential impacts on the water and energy cycles in tropics (Fan et al.,

2019). Further modifications of oil palm-specific canopy structure can help to understand the biophysical changes after oil palm conversion. Moreover, although the expansion of oil palm cultivation is seen as a severe threat for the conservation of rainforest and swamp areas and their associated ecosystem services (Koh and Wilcove, 2008; Koh et al., 2011), oil palm is admittedly the most productive oil crop with 3-5 times yields of other oil crops. To replace oil palm, much more lands will thus be needed for other oil crops to produce the same amount of oil production. This is also in dispute among policy-makers.

The model with explicit representation of oil palm and calibration using site-level data can provide spatial oil palm biomass density, yield and water consumption in future land use scenarios and would help to identify the most suitable areas for growing

oil palms as well as to contribute to the policy formulation for the sustainability of oil palm plantation, although the effects of soil carbon and nutrient content, and fertilization management on oil palm growth and yields still require further investigation.

**5 Conclusion**

In this study, oil palm was incorporated in the ORCHIDEE-MICT LSM as a new PFT by introducing the phytomer structure and a fruit harvest pool, modifying carbon allocation and implementing a systematic parameterization scheme. The leaf seasonality represented by different leaf age cohorts was also merged into this model. The developed MICT-OP version performs reasonably well in simulating photosynthesis, carbon allocation, biomass stock and fruit yields at multiple observation sites. Compared with the default ORCHIDEE-MICT version, ORCHIDEE-MICT-OP shows improved

performance of GPP partitioning, NPP allocation and biomass components. The new oil palm version, parameterized with age-specific parameters, generally captures temporal dynamics of oil palm biomass and yields. Implementation of more management practices (e.g., fertilization and irrigation) and parameterization of biophysical variables are further needed. Generally, our model improved the representation of oil palm in LSMs and further applications of ORCHIDEE-MICT-OP include but are not limited to regional carbon budget and water demand estimation, yield prediction and the sustainable

development of oil palm industry.

**Code availability**

The source code for ORCHIDEE-MICT-OP revision 6850 is available via https://forge.ipsl.jussieu.fr/orchidee/wiki/GroupActivities/CodeAvalaibilityPublication/ORCHIDEE-MICT-OP-r6850 (last access: 23 July 2020; Xu, 2020, the doi will be updated later). This software is governed by the CeCILL licence under French

law and abiding by the rules of distribution of free software. You can use, modify, and/or redistribute the software under the terms of the CeCILL licence as circulated by CEA, CNRS, and INRIA at the following URL: http://www.cecill.info.

**Competing interests.** The authors declare that they have no conflict of interest.

**Acknowledgements**

This study is supported by the National Key Research and Development Program of China (grant no. 2019YFA0606601,

2019YFA0606604 and 2017YFA0604401). Wei Li and Philippe Ciais acknowledge support by the European Research Council through Synergy Grant ERC-2013-SyG-610028 "IMBALANCE-P".

**Author contributions**

PC, LY and WL designed the project. YX developed the model code with help from WL, PC, XC, CY and HZ. YX wrote an initial draft of the manuscript. All authors participated in interpreting the results and refining the manuscript.

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

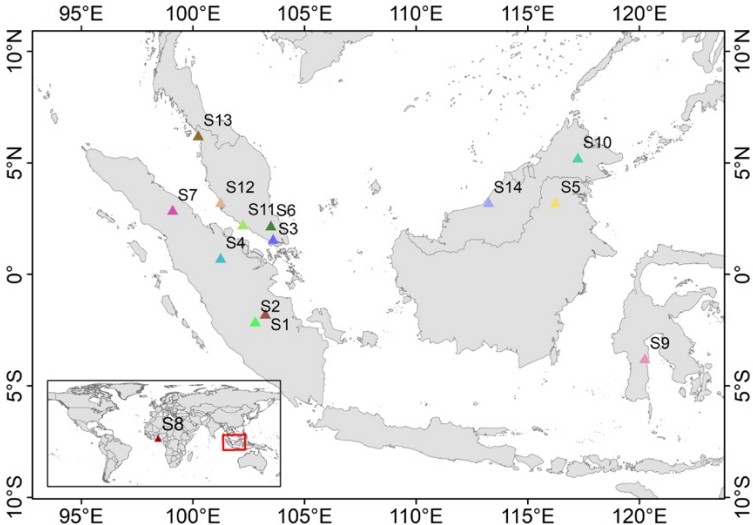

**Figure 1. Spatial distribution of the 14 observation sites used for model calibration and evaluation. The red rectangle in the inserted map shows the location of main map (Malaysia and Indonesia).**

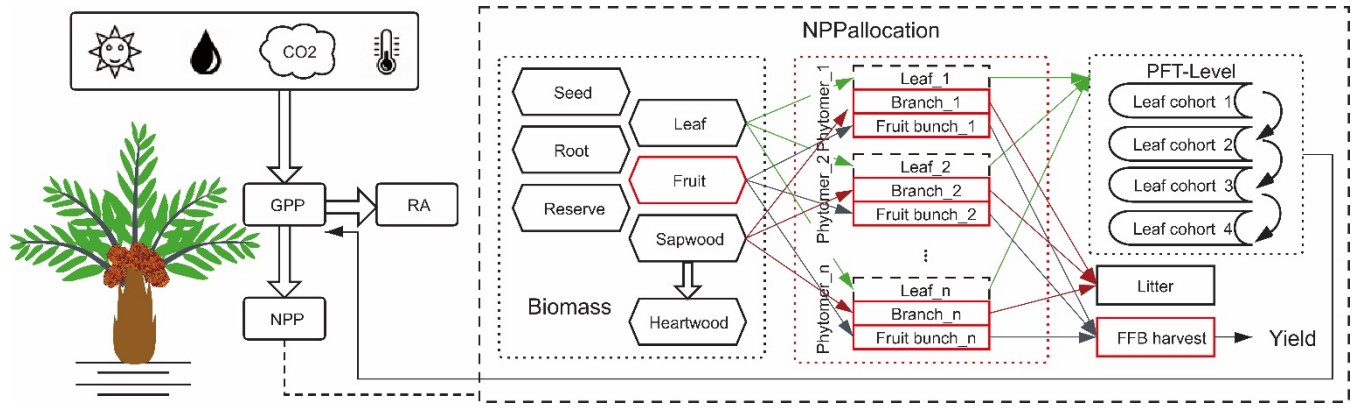

**Figure 2. Schematic diagram showing the implementation of oil palm in ORCHIDEE-MICT-OP. The major modifications and new plant organs / harvest module are highlighted using the red blocks. The branch and fruit components (solid lines) were implemented at the phytomer level, while leaf component (dashed lines) was simulated as a whole of all phytomers at the PFT level to remain consistent with the four leaf age cohorts of the modelled phenological equations. RA refers to the autotrophic respiration. FFB harvest refers to fresh fruit bunch harvest.**

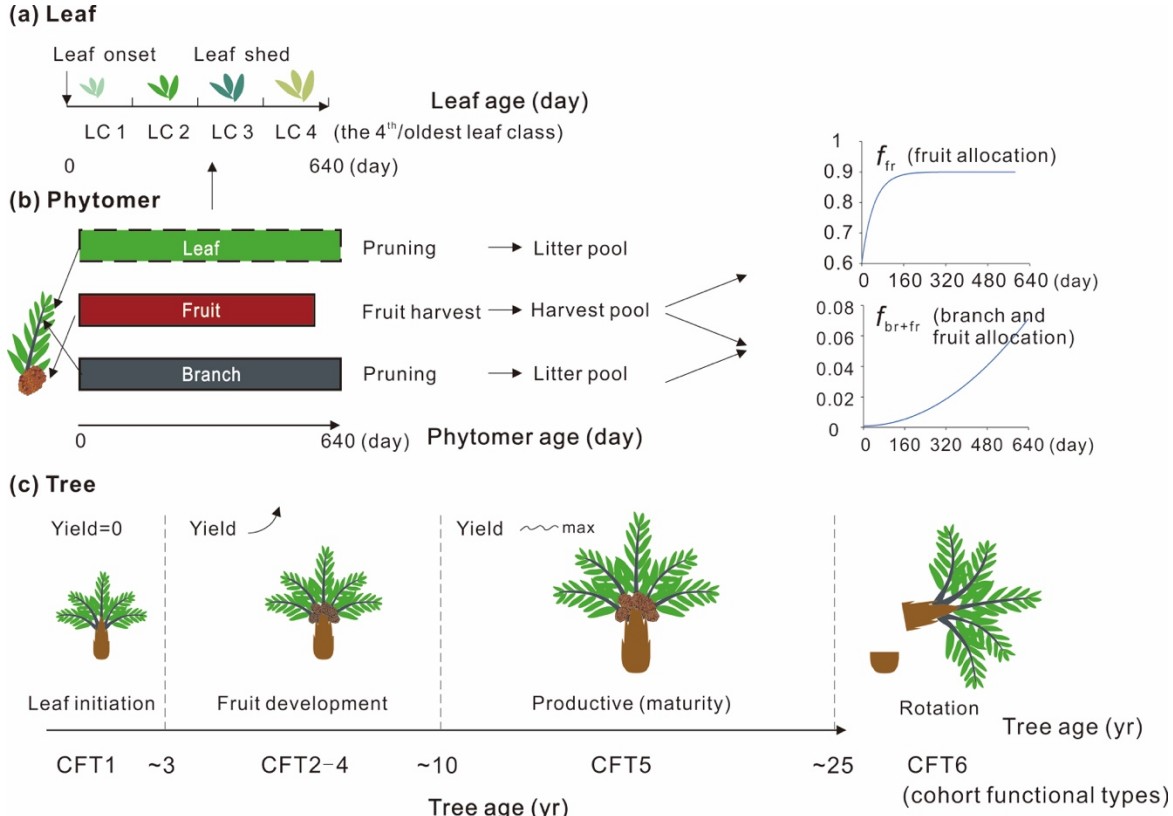

**Figure 3 Schematic of (a) leaf, (b) phytomer and (c) plant dynamics with leaf, phytomer and tree ages. The branch and fruit allocation is a function of phytomer age. The oil palm PFT experiences an increase of fruit yield during CFT 2-4 and reaches the maximum and steady yield at the most productive period (CFT5). The leaf component is not specifically simulated for each phytomer (dashed rectangle) but implemented at the PFT level with four leaf age cohorts. The major phenological phases for phytomer during the oil palm life cycle are presented with tree ages. LC and CFT refer to leaf cohort and cohort functional type, respectively.**

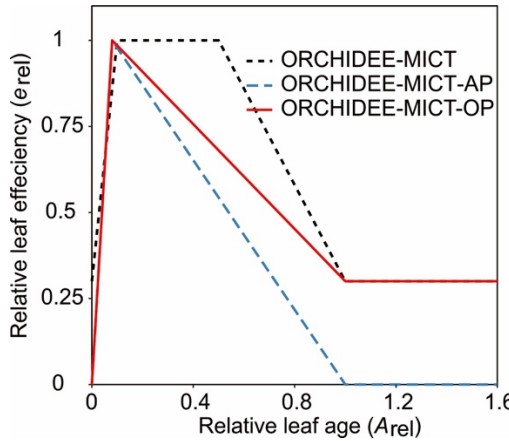

**Figure 4 Relative leaf efficiency (erel) as a function of relative leaf age (Arel) used in 1) this study, ORCHIDEE-MICT with oil palm (ORCHIDEE-MICT-OP), 2) the default ORCHIDEE-MICT version (ORCHIDEE-MICT) and 3) the ORCHIDEE-MICT version with the new leaf phenology scheme in Chen et al., 2019 (ORCHIDEE-MICT-AP).**

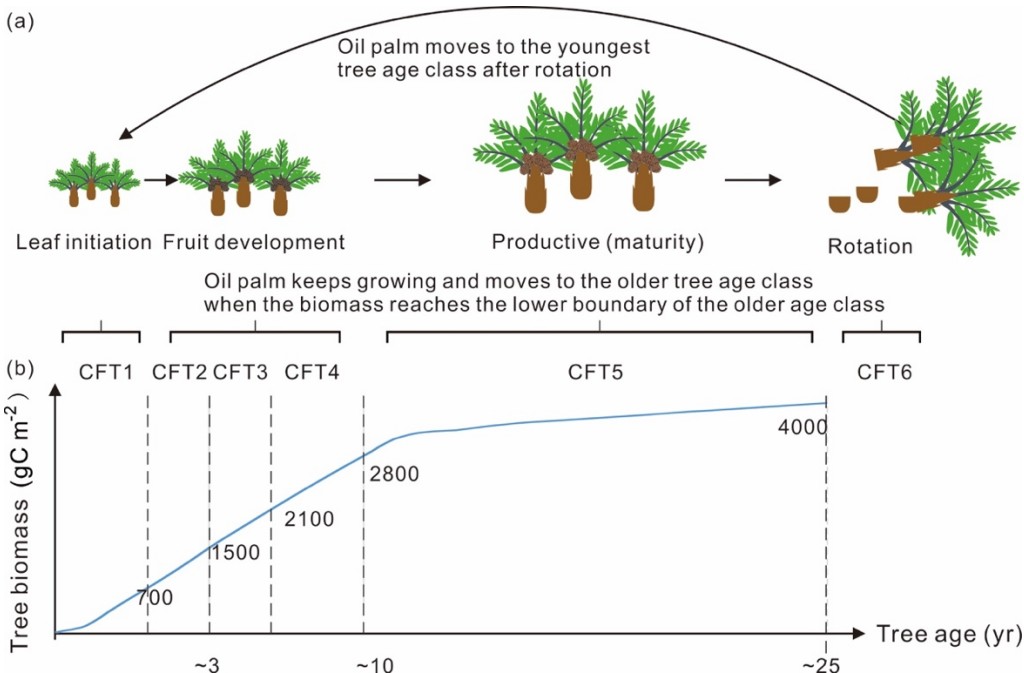

**Figure 5. Tree age classes of oil palm along with the temporal change of total biomass. a) an example of oil palm tree age class dynamics: 1) keep growing and move to the older tree age class; 2) move to the youngest age class after clear cutting for rotation. b) the growing curve of total biomass for oil palm tree. The labelled numbers are the biomass boundary of each CFT.**

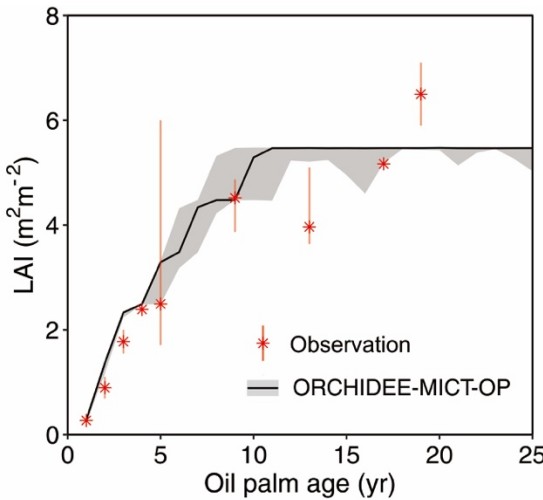

**Figure 6 Temporal dynamics of LAI for oil palm. The black solid line and the grey shade indicate the median and range of simulated LAI for oil palm across all sites in ORCHIDEE-MICT-OP. The error bars of observations represent the range of different observations at a certain age from various locations, treatments and species.**

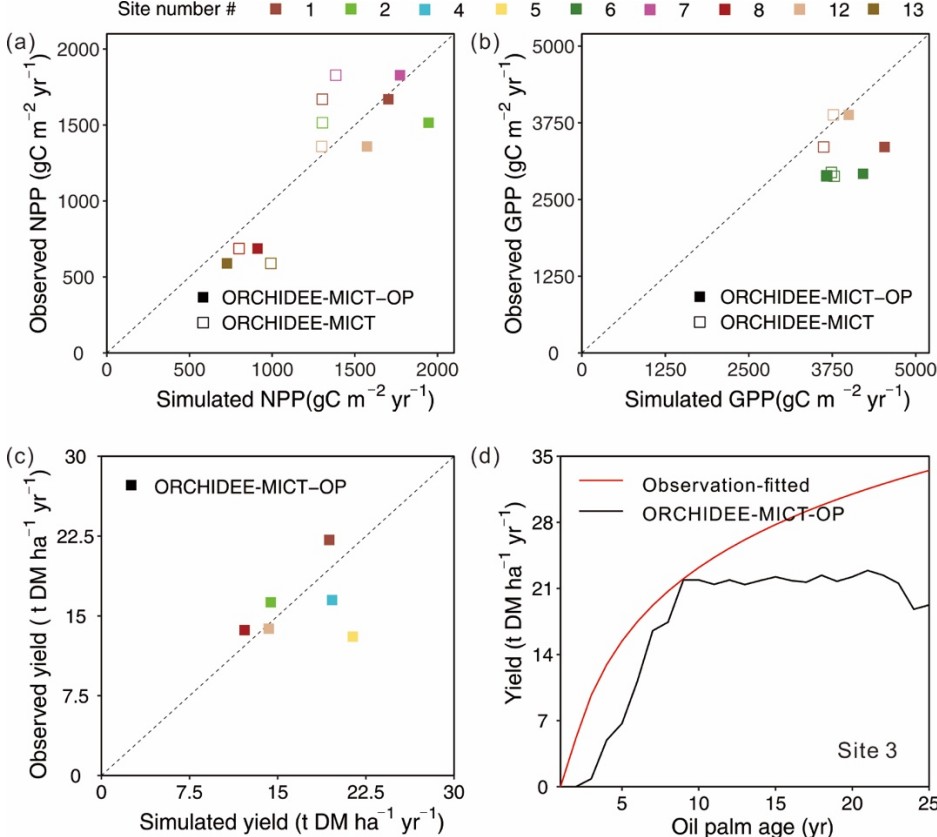

**Figure 7. Comparison of simulated (a) NPP, (b) GPP, (c) fruit yield and (d) temporal dynamics of yields against observations. "ORCHIDEE-MICT-OP" refers to the simulation results by the ORCHIDEE-MICT-OP version using the newly added oil palm PFT. "ORCHIDEE-MICT" refers to the simulation results by the default ORCHIDEE-MICT version using TBE tree PFT. The dashed line indicates the 1:1 ratio line.**

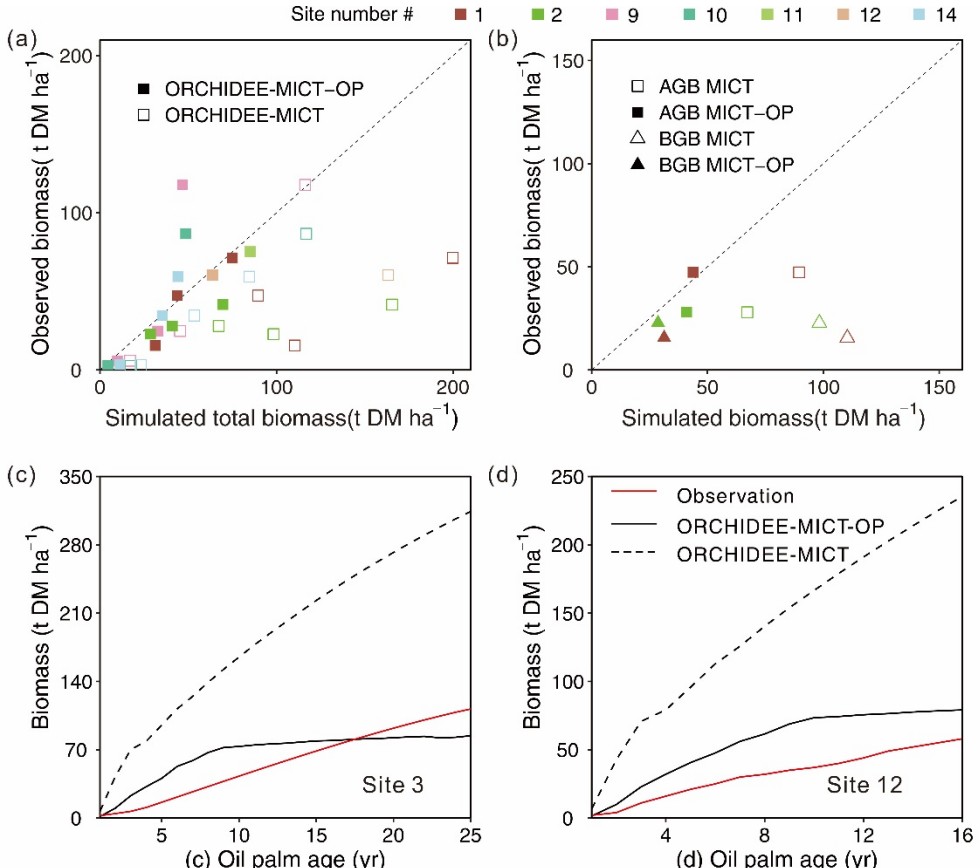

**Figure 8. Comparison of simulated (a) total biomass, (b) above ground biomass (AGB) and below ground biomass (BGB), temporal dynamics of estimated biomass for oil palm at (c) Site 3 and (d) Site 12 against observations. the observations from Site 3 and Site 12 were calculated by allometric equation using the measured diameter at breast height (DBH) and height of the stem. "ORCHIDEE-MICT-OP" refers to the simulation results by the ORCHIDEE-MICT-OP version using the newly added oil palm PFT. "ORCHIDEE-MICT" refers to the simulation results by the default ORCHIDEE-MICT version using TBE tree PFT. The dashed line in (a) and (b) indicates the 1:1 ratio line.**

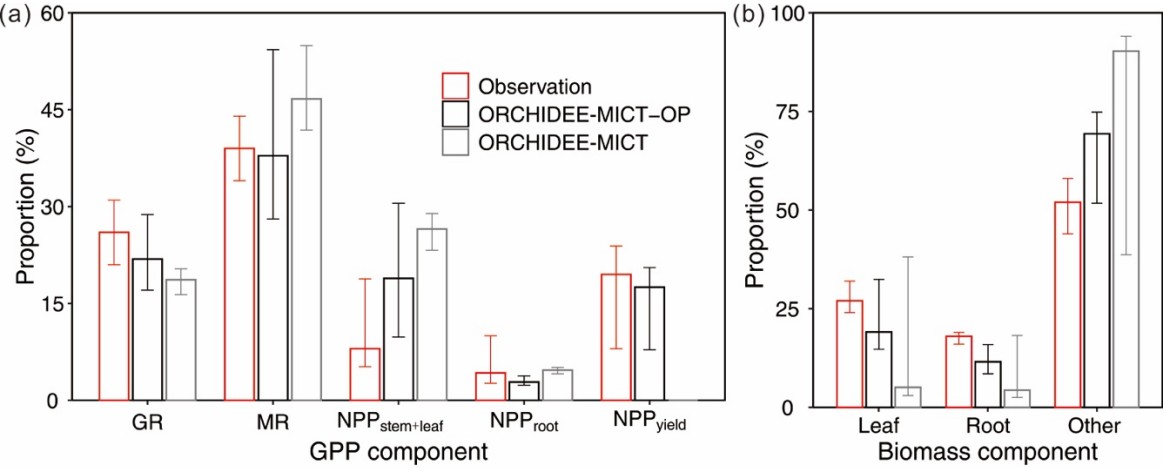

**Figure 9 Components of (a) GPP and (b) standing biomass. The fruit component in (b) is the developing fruit in the phytomer and the harvested fruit is not accounted in the total biomass. Error bars show the ranges across different sites and ages. GR and MR stand for growth respiration and maintenance respiration.**

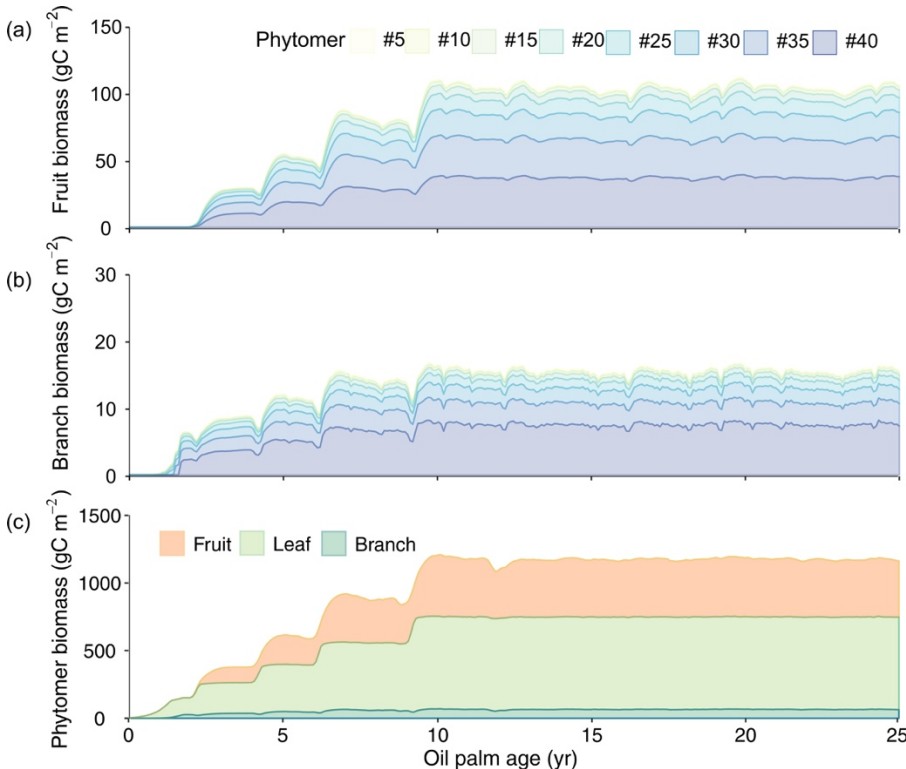

**Figure 10. Temporal development of phytomer biomass: (a) fruit (b) branch and (c) phytomer biomass. The colors in (a) and (b)**
**represent the fruits and branches from the eight representative phytomers. Only eight representative phytomers (#5, #10, #15, #20, #25, #30, #35 and #40) are shown in (a) and (b) for better visualization. The total phytomer biomass in (c) is split into fruit, leaf and branch biomass for all the 40 phytomers aggregated.**

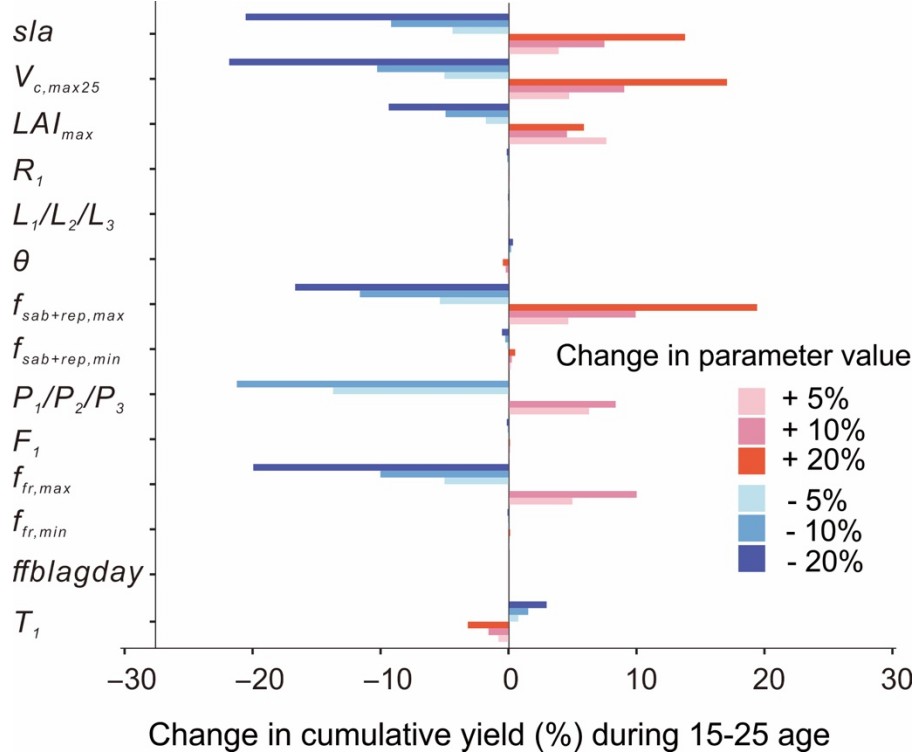

**Figure 11. Change in cumulative yields by varying ±5, ±10 and ±20% of the key parameters related to photosynthesis, allocation and turnover in the oil palm modelling. The parameter is changed one by one while the others are kept as the same.**