# Peer review of "Oil palm modelling in the global land-surface model ORCHIDEE-MICT"

_Geoscientific Model Development, 2020_

## Referee Comment (RC1) · Anonymous Referee #1 · 7 Jan 2021

Land use changes driven by oil palm expansions have been major concerns for carbon emissions and biodiversity loss in the tropics and have drawn extensive studies in the recent decade. Yet, modeling oil palm in a land surface scheme of Earth system model only started recently. Representing oil palm as a plant functional type (PFT) in an LSM with oil palm specific morphological, phenological and physiological traits including a sub-PFT structure for oil palm's phytomers was first introduced in CLM (CLM-Palm, Fan et al. 2015). Here, this study adopts a similar sub-PFT structure and presents some advances, such as an age-specific parameterization scheme for photosynthesis and autotrophic respiration for the oil palm PFT, which is new. ORCHIDEE-MICT has an age-cohort vegetative structure that is different from the CLM (excluding FATES), and the oil palm integration in this study was based on existing leaf age cohorts-based

phenology of tropical broadleaf trees and distinct age classes of the ORCHIDEE-MICT model. The developed model ORCHIDEE-MICT-OP shows reasonable agreement with observational data for simulating LAI, biomass pools, GPP and NPP. However, I recommend major revisions to address several main issues in the methodology as well as minor ones.

First, the methodology description of phenology and allocation needs to clarify how sub-PFT level processes are reconciled with the PFT-level processes in ORCHIDEE-MICT, particularly for leaf phenology. For example, it is unclear how the VPD-triggered leaf shedding for the whole palm works together with the phytomer-level leaf pruning. The allocation parameterization and results did not show sub-PFT/phytomer level leaf LAI or biomass dynamics. Without phytomer-specific leaf phenology, it is hard to call this a sub-PFT structure as individual leaf dynamics together with fruiting and harvest on each phytomer are the unique characteristic life cycle of oil palm (distinguishing it from natural trees). It is also difficult to understand the three-phase life cycle of the whole tree and the sub-PFT level phenology and allocation processes if without substantial clarification.

Second, the model calibration here involved published data from 14 individual sites for different variables of biomass, yield, LAI, GPP and NPP. However, there lacks an independent validation against separate sites or dataset. Calibration and validation should be conducted separately to ensure model generalizability and applicability. Moreover, several empirical parameters appear weakly constrained. Thus, a sensitivity analysis of newly introduced parameters is favourable but is currently missing. I also urge the authors to give more proper credit to the related work of CLM-Palm in several aspects and improve their model description and evaluation to highlight their own new/original contributions as mentioned above.

Below are specific comments:

L59: I suggest first referring to the CLM-Palm work here when talking about LSMs,

either put in parentheses like '…without a specific representation in LSMs (except CLM-Palm)', or mentioning it at the end of this paragraph that at least one LSM CLM4.5 already introduced an oil palm specific PFT and related parameterizations (see below comment). L74: better cite an observational study here, rather than a modeling study.

L78: when first mentioning ORCHIDEE, there needs a couple of sentences introducing this LSM, such that incorporating an oil palm PFT into ORCHIDEE would contribute to modeling the carbon, water and energy cycle of this perennial crop in a variety of LSMs, in addition to CLM.

L88-90: "using a sub-canopy framework from CLM4.5" is not a proper description. As far as I know, the original CLM4.5 does not have a sub-canopy or sub-PFT structure. This was introduced in Fan et al. (2015) to CLM4.5 specifically for oil palm. A proper citing of CLM-Palm here could be: CLM-Palm was the first LSM that introduced an oil palm specific PFT and a sub-canopy/sub-PFT framework for modelling oil palm's phytomer-based structure and phenological and physiological traits in CLM4.5, or something similar. There are other locations that need similar care.

L95: does "tree cutting" refer to pruning of old phytomers or the clear-cut at final rotation?

L95-97: as mentioned above, the authors should cite the CLM-Palm work here, because the "sub-PFT structure" was clearly defined in Fan et al. (2015), including carbon allocation for leaf and fruit of each phytomer and management practice of pruning, fruit harvest and rotation (see their Fig. 1, Fig. 2 and sections 2.1, 2.2).

L101: delete extra words. There are other typos in the text.

L115-120, Section 2.1: Although the number of sites used in this study seems abundant, the actual data availability for different variables is sparse at individual sites, e.g., only one Site-12 provided annual yield data and one Site-3 provided annual biomass data. This limits the model validation. It seems the author did not conduct independent model validation using new sites other than those already used for parameter calibration. Model calibration and model validation are different procedures to ensure applicability of a model to new locations. This section should at minimum describe what variables from which sites are used for calibration, and what variables from which other sites are used for independent validation.

L150-160, and Figure 3: It is hard to understand the phenology scheme from descriptions in section 2.3.2 and Figure 3. Please clarify if the oil palm will produce yield at CFT stages 1-4 (0-10 years), or only at CFT5 (10-25 years old)? From Figure 3c, it seems that fruit yield and harvest pertain only to CFT5 (productive stage), but from the phenology and allocation descriptions, it seems a phytomer will produce fruit as soon as its initiation and will be harvested before pruning when its age reach the longevity. The phytomer longevity is 640 days, which is smaller than 3 years of CFT1 – leaf initiation stage. Thus, the phytomer phenology implies fruiting and harvest starts around 2 years old (which is true according to field observations), but Figure 3c suggests otherwise (the CFT1 is only at leaf initiation stage with ÆŠfr=0 when palm age < 3 years). Great efforts are needed to clarify the logic link between PFT-level phenology and sub-PFT level phenology, especially how they are synchronized in the model?

L168: should be section 2.4?

L176-178: the described logic of phytomer initiation (controlled by management) and pruning (controlled by phytomer age) is counterintuitive. Here, phytomer initiation follows each pruning to maintain the total number of 40, but initiation should be controlled by physiological process rather than management. According to oil palm phenology and field management, phytomer initiation is regulated by the phyllochron (the thermal period between initiations of two successive phytomers) which increases with oil palm age, while pruning usually is done on the bottom phytomers (old but not necessarily dead ones) when managers observe the total number of phytomers exceeding ∼40. The described scheme of phytomer phenology is apparently weak, given the improper logic of initiation and pruning and use of fixed days for phytomer longevity.

L183-185: define SWdown; "weekly VPD is used to trigger the shedding of old leaves" – how the shedding of old leaves at the PFT level is merged with the above phytomer phenology? Each phytomer carries a large leaf, and it is initiated and pruned according to phytomer phenology. But the authors did not describe in detail how the phytomer-level leaf phenology is reconciled with the age-cohort based leaf phenology from Chen et al. (2020). Presumably they are synchronized, such that the sum of phytomer-level LAI and leaf biomass should always equal the PFT-level LAI and leaf biomass to maintain carbon balance. The authors mention earlier "the leaf, branch and fruit bunch belonging to each phytomer were linked with the original leaf, sapwood biomass pools," but from the schematic Figure 2, it is unclear how each leaf enters the litter pool. Does the leaf carbon enter the successive cohort cycle even after pruning of the supporting phytomer? At minimum, the authors should describe how the VPD-triggered leaf shedding works together with the phytomer-level leaf initiation and pruning in this section.

L188: "a fixed allocation of 10% to reproductive plant tissues" – how this fraction of reproductive allocation is related to the phytomer specific fruit allocation? Fruit is usually considered part of the reproductive organ.

L198: according to the calibrated values of coefficients ðİŚČ1, ðİŚČ2 and ðİŚČ3 , the maximum value of Eq. (1) is ÆŠbr+fr,min + (ÆŠbr+fr,max - ÆŠbr+fr,min) x 0.07 when a phytomer reaches max age; the range of the modifier 0 to 0.07 (instead of 0 to 1 normally) seems to be too small, which would exert very weak phenological effect on this allocation parameter.

Section 2.3.3: Inconsistency between model schematic in Figure 2 and allocation descriptions. In Figure 2, fruit to fruit-bunch allocation is independent of sapwood to branch allocation, but the descriptions and equations here suggest fruit allocation is part of sapwood allocation. Figure 2 also shows leaf to phytomer-leaf allocation, but there is no indication of this sub-PFT leaf allocation process in the text. Only a PFT-level allocation parameter ÆŠleaf is described in Eq. (5). If phytomer specific leaf

carbon pool is not modeled, some parts of Figure 2 should be modified. And the text description of sub-PFT phenology for branch, leaf and fruit for each phytomer is not accurate (e.g., L28 in abstract).

L219: "fruits will be harvested after the phytomer age in the oldest phytomer reaches ðİŘťgeffbcrit" – does this imply that fruit initiation happens at the same time as phytomer initiation? It needs to note that phytomers develop their leaves first before initiation of fruit. But the equations (1) to (5) suggest that fruit allocation at each phytomer starts immediately after phytomer initiation.

Section 2.4: this section lacks a sensitivity analysis for newly introduced oil palm parameters. Although the results show general agreement with observational data by a one-time calibration with all sites, readers gain little insight on how sensitive are the oil palm LAI or biomass pools to different parameters and to climate and surface forcing without showing a sensitivity analysis or independent validation using different sites.

L227: missing Yin and Struik, 2009 in the reference list.

Section 2.4.3: I suggest merging this section to section 2.3.3 as they both describe allocation. Again, leaf allocation (ÆŠleaf) is only described for the whole palm, not for each phytomer. If only an PFT integrated leaf carbon pool is grown for the whole palm, Figure 2 and several places in the text about sub-PFT structure should be modified.

Section 2.4.4: Since Ageleafcrit equals Agephycrit, one of these parameters could be eliminated. But it needs to note that, in reality the leaf longevity is smaller than phytomer longevity as there is a lengthy "spear leaf" (a bud that grows to as long as ~3 meters) stage before the it fully expands to be able to photosynthesize.

L300: if leaf carbon pool is not specifically simulated for each phytomer like branches and fruits, how pruning of each leaf is conducted, and how this is reconciled with the VPD-triggered leaf shedding? It needs to avoid double accounting of leaf litterfall flux from these two processes. In oil palm plantations, natural leaf loss/shedding is almost

impossible without pruning due to very high lignin content. Thus, I suggest only one leaf litterfall mechanism is implemented for oil palm.

L350-355: it will be more convincing to compare the simulated annual yield with real observations rather than the fitted curve in Figure 7d. This can be achieved either from using continuous harvest data or using space-for-time substitution with multiple sites.

L376-377: calibration an LSM using all sites without reserving some sites for independent validation is not standard. Also, the reason of overestimation should not be attributed to "no calibration was applied for this site". LSM is aimed for regional or global applications. Thus, even a perfect calibration for an individual site does not mean good predictive power for other sites. That's why I urge the authors to do independent validation after model calibration using separate datasets.

Section 3.5: again, the fact that leaf LAI and biomass pool is not specifically simulated for each phytomer should be clarified much earlier when introducing the sub-PFT structure and in Figure 2.

L412: "The fruit production and harvest begin after entering the fruit development phase" can be described earlier when introducing Figure 3. The 3-phase description is confusing as it sounds like only the productive phase (CFT5) has yield and harvest.

L440-441: Fan et al. implemented age-dependent carbon allocation strategy (their section 2.2.1) and showed age-dependent trend in yields validated against different sites (their Figure 10).

L445: this is not true; Fan et al. used 2 sites for model calibration and an additional 8 sites for independent validation. The study here used all 14 sites for both calibration and validation, which violates the normal procedure of model validation using independent sites and undermines its generalizability to other locations.

L446: this is not necessarily the case; parameterizations from ORCHIDEE-MICT-AP can only be applicable to other LSMs if they share very similar model structures. Without a sensitivity analysis and independent validation, such claim is premature.

Section 4.3: when talking about application in LUC impact assessment, it is important to note the limitation that effects of different land cover types, soil carbon and nutrient content, and fertilization management on oil palm growth and yield can hardly be represented by the current model without a nitrogen cycle.

––––––––––––––––––––

---

## Referee Comment (RC2) · Anonymous Referee #2 · 12 Jan 2021

General comment: This paper presents an improved, oil palm specific plant functional type (PFT) to be incorporated into the ORCHIDEE-MICT global land surface model . The authors offer this as an improvement on the existing typical practice of approximation using the PFT of tropical broadleaved evergreen trees. Given the intensively managed, and specific, nature of palm cultivation and harvesting there is little doubt as to how unsatisfactory the utilisation of a generic plant functional type is in modelling physiological and environmental dynamics in these systems. The paper is generally very well written and presented (with some caveats, see details below) and the results suggest a significant improvement over previous model practice. Reviewer #1 has provided some excellent and comprehensive technical comments and suggestions and I have little to add to those. I would comment though, that peat and mineral soil based

plantations are likely to be very different, yields will be lower and palm mortality higher on tropical peat soils and crop rotations significantly shorter (18 to 20 rather than 25 to 30 years). With around 25% of the South East Asian oil palm plantations being on converted peatlands, a discussion around the significant differences likely to be found in yield and palm mortality (disease incidence, palm root anchorage and failure) on the two soil types and whether they should be separately parameterised, and what work would be required to do so, in the PFT would be welcome.

Specific comments: ln.20: "cause" should be "causes" (plural)

ln.36: "crop" should be "crops" (plural)

ln.48: citations in the brackets should be in chronological order

ln.54-55: "do not allow to represent the land use change", poor English, perhaps: "do not allow representation of land-use change..."

ln.60-61: change "according to the genotypes and locations" to "dependent on genotype and locations..."

ln.84: rotations of 25-30 years would be specific to mineral soil-based plantations, peat-based plantations are likely to be significantly shorter

ln.148: change "corresponded" to "corresponding"

ln. 158: "begin to flourish"

ln.201: a space needed "accelerated (Corley and Tinker, 2015)

ln. 222: delete "types", already included in the PFT acronym

ln.241: change "correspondingly" to "corresponding"

ln.305: "we assumed that there is no natural mortality for the oil palm,", this is unrealistic as there will be disease losses from the original planting density (with some replacement) and specifically in peatlands, losses and yield reduction due to leaning

palms. These need discussing.

ln.313: "Figure S3" should be "Figure 3"

ln.369: "is reproduced of 1.7" Perhaps this should be "is calculated at..."?

ln.376: this distinction between peat and mineral-based plantations, and its implication for scaling model output needs to be discussed further

ln.408: change "characteristics" to "characteristic" (singular)

ln.436: change "special" to "specific"

ln.440: change "increasing" to "increase"

ln.440-441: "To our best knowledge, age-based allocation dynamics for oil palm have not yet been..."

ln.441: change "simulating" to "simulate"

ln.452: change "phytomer" to "phytomers" (plural)

ln.487-488: the authors mention the impact that planting density will have when scaling from palm to area, again this will differ between peatland and mineral and needs more discussion.

ln.496-497: The authors state that "soil carbon change may take a long time", they do not elaborate on what a "long time" is, but in peat systems there are huge carbon emissions observed in the first 5 years following conversion, likely a similar amount of time needed for forest residues to decompose which would disagree with the thrust of this sentence.

ln. 499-500: change ",it is also in urgency to..." to ",there is an urgent need to..."

ln.504: change "crops" to "crop" (singlular)

Figure 3: change "prunning" to "pruning". Also, please include full versions of acronyms

[Figure]

used in the figures in their associated legends (e.g. CFT in fig.3). The legend for fig. 3 is generally inadequate and needs more detail. It should stand alone and not require recourse back to the text.

---

## Author Comment (AC1) · 5 Apr 2021

*Referee #1*

**[General comment]** *Land use changes driven by oil palm expansions have been major concerns for carbon emissions and biodiversity loss in the tropics and have drawn extensive studies in the recent decade. Yet, modeling oil palm in a land surface scheme of Earth system model only started recently. Representing oil palm as a plant functional type (PFT) in an LSM with oil palm specific morphological, phenological and physiological traits including a sub-PFT structure for oil palm's phytomers was first introduced in CLM (CLM-Palm, Fan et al. 2015). Here, this study adopts a similar sub-PFT structure and presents some advances, such as an age-specific parameterization scheme for photosynthesis and autotrophic respiration for the oil palm PFT, which is new. ORCHIDEE-MICT has an age-cohort vegetative structure that is different from the CLM (excluding FATES), and the oil palm integration in this study was based on existing leaf age cohorts-based phenology of tropical broadleaf trees and distinct age classes of the ORCHIDEE-MICT model. The developed model ORCHIDEE-MICT-OP shows reasonable agreement with observational data for simulating LAI, biomass pools, GPP and NPP. However, I recommend major revisions to address several main issues in the methodology as well as minor ones.*

**[Response]** We thank the reviewer for the careful review and helpful comments and suggestions. We have revised the manuscript according to your suggestions, and we believe the model structure and validation have been clarified in the revised manuscript. Please see the detailed point-by-point responses below.

**[General comment]** *First, the methodology description of phenology and allocation needs to clarify how sub-PFT level processes are reconciled with the PFT-level processes in ORCHIDEE- MICT, particularly for leaf phenology. For example, it is unclear how the VPD-triggered leaf shedding for the whole palm works together with the phytomer-level leaf pruning. The allocation parameterization and results did not show sub-PFT/phytomer level leaf LAI or biomass dynamics. Without phytomer-specific leaf phenology, it is hard to call this a sub-PFT structure as individual leaf dynamics together with fruiting and harvest on each phytomer are the unique characteristic life cycle of oil palm (distinguishing it from natural trees). It is also difficult to understand the three-phase life cycle of the whole tree and the sub-PFT level phenology and allocation processes if without substantial clarification.*

**[Response]** To address the reviewer's concerns on how the sub-PFT level processes are reconciled with the PFT-level processes in regarding of leaf shedding and phytomer pruning, we clarified that the leaf dynamics was implemented as a whole with 4 leaf age cohorts at the PFT level as a simplification, i.e. not at phytomer level (Section 2.3.1, Lines 150-152). These four leaf age cohorts were already implemented in the model to capture the phenology of forests given a total leaf longevity parameter that defines the maximum age of a cohort, it would be too complex to further divide leaf cohorts at the phytomer level (the dimension = 4 cohorts × 40 phytomers = 160). Since the photosynthesis was calculated at the PFT-level using the sum of all leaf biomass in each leaf age cohort, and the leaf biomass is relatively small compared to the branches and fruit biomass, we used PFT-level leaf cohorts. Although the phytomer level leaf LAI and biomass were not specifically simulated, the carbon allocation, pruning and harvest for branch and fruit component for each phytomer were implemented in our model with sub-PFT dynamics. Meanwhile, we also linked the growth of phytomer-level branch and fruit to the PFT-level changes using an age-specific parameterization. For the leaf shedding scheme, leaf longevity used in the VPD triggered leaf shedding scheme of Chen et al. (2020) is first modified as being the same as phytomer longevity (640 days) to approximate the natural shedding of old leaves in phytomer. Considering that the removal of leaves is not very well represented at the time of phytomer pruning, we further added an extra leaf turnover of the old leaves (using the leaf age cohort) at the time when the oldest phytomer is manually pruned. We modified and clarified the model structure and leaf phenology in all the related text through the manuscript. For the details, please refer to the reply to #9, #12 and #21.

We added more explanations on the three-phase life cycle of the tree in several places of the manuscript and modified Figure 3 to avoid misunderstanding, "The first phase corresponded to CFT1 is the first two years between oil palm establishing and the beginning of fruit-fill. In this period, leaf and branch begin to flourish and expand without fruit production. The second phase (corresponded to CFT2-4) is the fruit development phase when fruit begins to grow and harvest begins, while fruit and branch biomass continue to increase. The third phase corresponded to CFT5 is the productive phase with high and stable yields that will last until

the age of 25-30 years old." (Please refer to Lines 168-172 in the Section 2.3.2 and reply to #9). For the sub-PFT level phenology and allocation processes, please see the details of phytomer initiation in the reply to #11, phytomer and fruit allocation in reply to #13-15, leaf shedding in reply to #12 and #21 and fruit harvest in reply to #16.

**[General comment]** *Second, the model calibration here involved published data from 14 individual sites for different variables of biomass, yield, LAI, GPP and NPP. However, there lacks an independent validation against separate sites or dataset. Calibration and validation should be conducted separately to ensure model generalizability and applicability. Moreover, several empirical parameters appear weakly constrained. Thus, a sensitivity analysis of newly introduced parameters is favorable but is currently missing. I also urge the authors to give more proper credit to the related work of CLM-Palm in several aspects and improve their model description and evaluation to highlight their own new/original contributions as mentioned above.*

**[Response]** We agree with the reviewer that it is better to separate calibration and validation sites. However, it is difficult to obtain enough continuous observations in one or two sites to constrain the model from public available data. In reality, most published studies provided limited observations (1 or 2 variables) at one time phase, which is not enough to track the growth of oil palm. Due to the lack of accessible observations, we have to utilize the existing knowledge of oil palm growth phenology and plantation management, together with the range of field observations from the sites to constrain the model. We admitted the limitations of the method because of the shortage of observations. With more field observations become available in the future, we believe it would be of great help for the modelling community, and we will also benefit from further calibration and validation of our model.

As suggested, we added a test by recalibrating the model using one site with most observations compared to other sites, and we then validated the model using the remaining sites in the supplement (Figure S5). We compared the new results (ORCHIDEE-MICT-OPv2) with the observations as well as the original results (ORCHIDEE-MICT-OP, calibrated using information from all the sites). The overall pattern of the simulation results was similar between the two parameterization schemes, and both showed a great improvement from the default PFT2 (ORCHIDEE-MICT) in biomass, GPP, NPP and yield, with some sites closer to observations and others more biased (Figure S5). Facing the difficulty in acquiring the original observation records for independent sites, we made a similar figure to previous studies (Figure 6 in Fan et al., 2015 and Figure 11 in Adachi et al., 2018, reproduced below) using the same style and scales to visually compare the temporal dynamics of simulated yields. The simulated annual and cumulative yields showed good agreement with observations in the two sites, indicating the model's ability to capture yield dynamics. For the details of the model calibration and validation, please refer to the reply to #8 and #22.

As for the concerns on the model parameterization. we added a set of sensitivity tests in Section 2.4.5 and Section 3.6. We tested the major carbon allocation parameters and the corresponding changes of cumulative yields. The results also indicate that yields are sensitive to parameters such $V_{cmax25}$ and phytomer allocation coefficient ($P_1/P_2/P_3$)) but less sensitive to the other parameters (such as $f_{fr,min}$, $F_1$). For the details, please refer to the reply to #17.

In addition, as suggested, we added credits to pioneer work of CLM-Palm in modeling oil palm in LSMs (Please refer to Lines 77-79, 89-90, 93-95, 98-100 in the Section 1 and Response to #1, #3, and #4).

All of the specific comments and suggestions have been addressed and implemented in the revised manuscript. Please find below the specific reviewer's comments, followed by our responses and relevant changes in the manuscript. We think that the revised version addresses all the issues raised by the reviewer.

**Below are specific comments:**

**[Comment 1]** *L59: I suggest first referring to the CLM-Palm work here when talking about LSMs, either put in parentheses like '...without a specific representation in LSMs (except CLM-Palm)', or mentioning it at the end of this paragraph that at least one LSM CLM4.5 already introduced an oil palm specific PFT and related parameterizations (see below comment).*

**[Response to #1]** Thanks for this comment. We emphasis the work of CLM-Palm in parentheses as suggested, "Vegetation in most LSMs is represented by a discrete number of plant functional types (PFTs) and oil palm is approximated by tropical broadleaved evergreen (TBE) trees without a specific representation in LSMs (except CLM-Palm), although the physiological characteristics of oil palm differ from generic TBE trees." Please see Section 1, Lines 57-60 in this revision.

**[Comment 2]** *L74: better cite an observational study here, rather than a modeling study.*

**[Response to #2]** We changed the citation to the observational study here and modified the description. "Fruit bunches are developed in the axil of each phytomer and each phytomer experiences a life cycle from leaf initiation, inflorescences and fruit developing to harvest and pruning (Corley and Tinker, 2015; Lewis et al., 2020)." Please see Section 1, Lines 73-75 in this revision.

*Corley, R. H. V., and Tinker, P. B.: The oil palm, 5th ed., John Wiley & Sons, 2015.*
*Lewis, K., Rumpang, E., Kho, L. K., McCalmont, J., Teh, Y. A., Gallego-Sala, A., and Hill, T. C.: An assessment of oil palm plantation aboveground biomass stocks on tropical peat using destructive and non-destructive methods, Scientific Reports, 10, 2230, 10.1038/s41598-020-58982-9, 2020.*

**[Comment 3]** *L78: when first mentioning ORCHIDEE, there needs a couple of sentences introducing this LSM, such that incorporating an oil palm PFT into ORCHIDEE would contribute to modeling the carbon, water and energy cycle of this perennial crop in a variety of LSMs, in addition to CLM.*

**[Response to #3]** As suggested, we changed the sentences to "Currently, the biomass pool of phytomers is not included in the generic tree PFTs of most land surface models except CLM-Palm, which prevents us from modelling phytomer-specific development, monthly harvest and pruning." (please see Section 1, Lines 77-79), and we further emphasized the importance of this work in the next paragraph "Incorporating an oil palm PFT into ORCHIDEE would contribute to modeling the carbon, water and energy cycle of this perennial crop in a variety of LSMs except for CLM that already implemented oil palm modelling." (Please see Section 1, Lines 93-95). More details about ORCHIDEE can be found in the first paragraph of Section 2.2.

**[Comment 4]** *L88-90: "using a sub-canopy framework from CLM4.5" is not a proper description. As far as I know, the original CLM4.5 does not have a sub-canopy or sub-PFT structure. This was introduced in Fan et al. (2015) to CLM4.5 specifically for oil palm. A proper citing of CLM-Palm here could be: CLM-Palm was the first LSM that introduced an oil palm specific PFT and a sub-canopy/sub-PFT framework for modelling oil palm's phytomer-based structure and phenological and physiological traits in CLM4.5, or something similar. There are other locations that need similar care.*

**[Response to #4]** As suggested, we changed the sentences here and other places in the manuscript.

**[Comment 5]** *L95: does "tree cutting" refer to pruning of old phytomers or the clear-cut at final rotation?*

**[Response to #5]** The tree cutting here refers to the clear-cut of oil palm PFT at final rotation. We modified the texts here to "The oil palm growth from leaf initiation, fruit development, maturity to the clear-cutting of oil palm PFT at rotation were represented in the ORCHIDEE LSM." to avoid misunderstanding (see Section 1, Lines 97-98 in this revision).

**[Comment 6]** *L95-97: as mentioned above, the authors should cite the CLM-Palm work here, be- cause the "sub-PFT structure" was clearly defined in Fan et al. (2015), including car- bon allocation for leaf and fruit of each phytomer and management practice of pruning, fruit harvest and rotation (see their Fig. 1, Fig. 2 and sections 2.1, 2.2).*

**[Response to #6]** We added the citation of CLM-Palm here as "A sub-PFT structure—phytomer with branch and fruit (leaf component was implemented at PFT-level with four leaf age cohorts) for oil palm was implemented in ORCHIDEE based on the sub-PFT structure incorporated in the CLM-Palm (Fan et al., 2015)." (please see Section 1, Lines 98-100)

**[Comment 7]** *L101: delete extra words. There are other typos in the text.*

**[Response to #7]** Revised accordingly.

**[Comment 8]** *L115-120, Section 2.1: Although the number of sites used in this study seems abundant, the actual data availability for different variables is sparse at individual sites, e.g., only one Site-12 provided annual yield data and one Site-3 provided annual biomass data. This limits the model validation. It seems the author did not conduct independent model validation using new sites other than those already used for parameter calibration. Model calibration and model validation are different procedures to ensure applicability of a model to new locations. This section should at minimum describe what variables from which sites are used for calibration, and what variables from which other sites are used for independent validation.*

**[Response to #8]** For the concern of lacking comparison with continuous observations, we added a visual comparison with the observations in the figure of publications (reproduced below), due to the inaccessibility of the original data. Specifically, we compared the temporal dynamics of simulated yields with the observations from age 0 to age 20 of Figure 11 in Teh and Cheah 2018 and the cumulative harvest records from age 0 to age 13 in Figure 6 of Fan's paper (Fan et al., 2015). Since none of these two sites were used in the model calibration, their yields can be taken as an independent validation. The simulated annual and cumulative yields showed good agreement with observations in the two sites, indicating the model's ability to capture yield dynamics. The reproduced yield dynamics were added in the supplement (Figure S6 and Figure S7) and Lines129-131 and Lines 392-395.

[Figure]

(a) Figure from Teh and Cheah (2018)

(b) Our simulation results

**Figure Comparison of the temporal dynamics of yields against observations from the Merlimau estate, Melaka (2.25°N, 102.45°E). (a) Model results (derived from an oil palm growth and yield model, PySawit) and observations was duplicated from Figure 11 in Teh and Cheah (2018). (b) Simulated results using ORCHIDEE-MICT-OP following the figure style of (a). In (a), the oil palm plantations were planted at following density of 120, 135, 148, 164, 181, 199, 220, 243, 268 and 296 palms ha$^{-1}$ and the yields were given at the corresponding planting densities. YAP is year after planting. Values in brackets denote NMBE, NMAE and dr, where NMBE is the normalized mean bias error, NMAE is the normalized mean absolute error and dr is the revised index of agreement.**

[Figure]

**Figure Comparison of the simulated cumulative yield and harvest data (2005–2014) from the Site PTPN-VI in Jambi, Sumatra (1°41.6′ S, 103°23.5′ E). The left figure are the observation and calibration results in Site 1 duplicated from Figure 6 in Fan et al., 2015, while the right figure are the simulated results at the same site using ORCHIDEE-MICT-OP.**

For the concern of model applicability, we agree with the reviewer that it is better to separate calibration and validation sites. However, it is difficult to obtain enough continuous observations in one or two sites to constrain the model from the public available data. In reality, most published studies provided limited observations (1 or 2 variables) at one time phase, which is not enough to track the growth of oil palm. Also notice there is inconsistency in some records. For example, the yields in Site 3 are continuously increased with age, whereas in other sites (Site 12) and existing literatures (Corley and Tinker, 2015; Teh and Cheah, 2018), the oil palm yields will reach the maximum volume and keep steady after 6-10 ages. Therefore, we have to utilize the existing knowledge of oil palm growth phenology and plantation management, together with the range of field observations from the collected sites to constrain the model. We admitted the limitations of the method because of the shortage of observations. With more field observations become available in the future, we believe it would be of great help for the modelling community, and we will also benefit from further calibration and validation of our model.

As suggested, we added a test by recalibrating the model using the site with most observations compared to other sites, and we then validated the model using the remaining sites in the supplement (please see Section 2.1, Lines 125-129 and Figure S4, S5 in the supplement). Here, we calibrated our model using the LAI, yield, Biomass, and NPP partitioning from Site #12 (Figure S4) and validated the model using the rest of independent sites (Figure S5). In site 12, these observations were given at figure without exact observation values for oil palm planting density of 120, 160 and 200 palm ha$^{-1}$ (the corresponding observation ranges were given in red shades in the figure below).

[Figure]

**Figure S4. Comparison of model simulated (a) LAI and (b) yield dynamics with field measurements in Site 12 used for calibration (ORCHIDEE-MICT-OPv2). The red ranges refer to the given results for different oil palm planting densities varying from 120-200 palm ha$^{-1}$.**

We then validated the modelled GPP, NPP, yield and biomass at the remaining independent sites. By comparing the new results (ORCHIDEE-MICT-OPv2), the previous simulations (ORCHIDEE-MICT-OP) and the default PFT2 simulation (ORCHIDEE-MICT) with the observations, we found the overall pattern of the simulation results was similar between the two parameterization schemes, and both showed great improvement from the default PFT2 (ORCHIDEE-MICT) in biomass, GPP, NPP and yield, with some sites closer to observations and others more biased (Figure S5).

[Figure]

**Figure S5. Comparison of simulated (a) NPP, (b) GPP, (c) fruit yield, (d) total biomass, (e) above ground biomass (AGB) and below ground biomass (BGB), temporal dynamics of estimated biomass for oil palm at (f) Site 3. "ORCHIDEE-MICT-OP" refers to the simulation results by the ORCHIDEE-MICT-OP using the newly oil palm PFT and the calibration scheme using all the 14 sites. "ORCHIDEE-MICT-OPv2" refers to the simulation results using independent calibration and independent validation sites. "ORCHIDEE-MICT" refers to the simulation results by the default ORCHIDEE-MICT version using TBE tree PFT. The dashed line indicates the 1:1 ratio line. The overall pattern of the simulation results was similar in the two calibration schemes and both showed great improvement compared with the default PFT2 (ORCHIDEE-MICT) version. The simulated total biomass, AGB and BGB were all similar in the two calibration schemes. Simulated NPP by the independent validation scheme is closer to observation while GPP and yields are more or less biased compared with the original scheme.**

**[Comment 9]** *L150-160, and Figure 3: It is hard to understand the phenology scheme from descriptions in section 2.3.2 and Figure 3. Please clarify if the oil palm will produce yield at CFT stages 1-4 (0-10 years), or only at CFT5 (10-25 years old)? From Figure 3c, it seems that fruit yield and harvest pertain only to CFT5 (productive stage), but from the phenology and allocation descriptions, it seems a phytomer will produce fruit as soon as its initiation and will be harvested before pruning when its age reach the longevity. The phytomer longevity is 640 days, which is smaller than 3 years of CFT1 – leaf initiation stage. Thus, the phytomer phenology implies fruiting and harvest starts around 2 years old (which is true according to field observations), but Figure 3c suggests otherwise (the CFT1 is only at leaf initiation stage with $f_{fr,max} = 0$ when palm age < 3 years). Great efforts are needed to clarify the logic link between PFT-level phenology and sub- PFT level phenology, especially how they are synchronized in the model?*

**[Response to #9]** In our model, the oil palm PFT will start to produce yield at CFT2 and the fruit yields will increase from CFT2 to CFT4 and reach the maximum at CFT5 (the most productive period). We added several sentences to explain Figure 3c and modified the captions to make them clear (please see Figure 3 below). The sentences in the phenology and allocation part were also modified to clearly state that the fruit yield starts from CFT2, "The first phase is the first two years between oil palm planting and the beginning of fruit-fill. In this period, leaf and branch begin to flourish and expand without fruit production. The second phase (corresponded to CFT2-4) is the fruit development phase when fruit begins to grow and harvest, while fruit and branch biomass continue to increase." (Please refer to Lines 168-171 in the Section 2.3.2), "Here, the first phase of oil palm growth from age 0-2 is corresponding to CFT1, and the second phase corresponding to CFT2-4 starts from the end of age 2. The most productive phase is corresponding to CFT5 from age ~10-25 (Figure 3)" (Section 2.3.2, Lines 178-180 and Figure 3). Detailed parameterization for the new oil palm CFTs is presented in Section 2.4.

Fruit production starts from the end of the second year, which is synchronized with the phytomer longevity (640 days). The first phase (CFT1) is the first two years. In our model, we merged the oil palm sub-PFT level phenology with the framework of forest age classes in ORCHIDEE-MICT by using age-specific controls and parameterizations. In ORCHIDEE-MICT-OP, the oil palm PFT is further divided into 6 age cohorts (CFT). When the oil palm PFT bounces to the older CFT, the variables (including phytomer related variables) will be inherited to the older CFT with varying age-specific settings. For example, when the oil palm jumps from CFT 1 to the CFT2, the phytomer related variables will be inherited and more carbon will be allocated to fruit than leaf with the age-specific parameterization scheme. When the oil palm PFT turns to CFT6, the oil palm tree will be clear-cut using the oil palm rotation module. The advantage of combining the tree age cohort in the implementation of oil palm PFT is that the model can better track the carbon, water, energy changes induced by oil palm-related gross land use cover changes.

[Figure]

**Figure 3 Schematic of (a) leaf, (b) phytomer and (c) plant dynamics with leaf, phytomer and tree ages. The branch and fruit allocation is a function of phytomer age.** **The oil palm PFT experiences an increase of fruit yield during CFT 2-4 and reaches the maximum and steady yield at the most productive period (CFT5). The leaf component is not specifically simulated for each phytomer (dashed rectangle) but implemented at the PFT level with four leaf age cohorts.** **The major phenological phases for phytomer during the oil palm life cycle are presented with tree ages.** **LC and CFT refer to leaf cohort and cohort functional type, respectively.**

**[Comment 10]** *L168: should be section 2.4?*

**[Response to #10]** Yes, we changed it to Section 2.4.

**[Comment 11]** *L176-178: the described logic of phytomer initiation (controlled by management) and pruning (controlled by phytomer age) is counterintuitive. Here, phytomer initiation follows each pruning to maintain the total number of 40, but initiation should be controlled by physiological process rather than management. According to oil palm phenology and field management, phytomer initiation is regulated by the phyllochron (the thermal period between initiations of two successive phytomers) which increases with oil palm age, while pruning usually is done on the bottom phytomers (old but not necessarily dead ones) when managers observe the total number of phytomers exceeding ~40. The described scheme of phytomer phenology is apparently weak, given the improper logic of initiation and pruning and use of fixed days for phytomer longevity.*

**[Response to #11]** According to the field observations, the average temperature of the coldest month of the year for oil palm growth should not fall below 15 °C, and the optimal temperature condition ranges between 24 and 28 °C (Corley and Tinker, 2015). Oil palm stomata began to close when air temperature rose above 32°C (Rees 1961). In the main oil palm growing areas, temperatures are relatively uniform throughout the year (fluctuated at ~27°C) and rarely falling below 22°C (see the monthly temperature variations below). Therefore, GDD and low temperature may not be the major limitations for oil palm growth. In addition, regular harvest and pruning practice is conducted in the commercial oil palm plantations, which regulates the total number of phytomers. This assumption in our model is thus a balance between the plant growth and human management practices. The simplification/compromise and its weakness were also added and discussed in Section 4.3, Lines 516-524. We agree with reviewer that it is more reasonable to regulate the phytomer initiation by the phyllochron, but it will need substantial coding development considering the different model structure of ORCHIDEE from CLM. We will consider this point in future model developing work.

*Corley, R. H. V., and Tinker, P. B.: The oil palm, 5th ed., John Wiley & Sons, 2015.*
*Rees, A. R.: Midday Closure of Stomata in the Oil Palm Elaeis guineensis. Jacq, J. Exp. Bot., 12, 129-146, 10.1093/jxb/12.1.129, 1961.*

[Figure]

**Figure S9 Seasonal temperature variations over the global oil palm plantation area during the past 30 years (1986-2015). The red solid red line and the shade indicate the median and range of seasonal temperature variations derived from the global oil palm plantation map (dataset from Cheng et al., 2018). The temperature was based on the climate data from the CRUNCEP gridded dataset (Viovy, 2011) and averaged by month.**

**[Comment 12]** *L183-185: define SWdown; "weekly VPD is used to trigger the shedding of old leaves" – how the shedding of old leaves at the PFT level is merged with the above phytomer phenology? Each phytomer carries a large leaf, and it is initiated and pruned according to phytomer phenology. But the authors did not describe in detail how the phytomer- level leaf phenology is reconciled with the age-cohort based leaf phenology from Chen et al. (2020). Presumably they are synchronized, such that the sum of phytomer- level LAI and leaf biomass should always equal the PFT-level LAI and leaf biomass to maintain carbon balance. The authors mention earlier "the leaf, branch and fruit bunch belonging to each phytomer were linked with the original leaf, sapwood biomass pools," but from the schematic Figure 2, it is unclear how each leaf enters the litter pool. Does the leaf carbon enter the successive cohort cycle even after pruning of the supporting phytomer? At minimum, the authors should describe how the VPD-triggered leaf shedding works together with the phytomer-level leaf initiation and pruning in this section.*

**[Response to #12]** We defined shortwave downwelling radiation ($SW_{down}$) in Section 2.3.2, Line 196 in the revised manuscript. In reality, the phytomer has three components-leaf, branch and fruit bunch. In our model structure, we implemented 4 leaf age classes with $VPD$ and $SW_{down}$ triggered leaf shedding and leaf flourishing scheme from Chen et al 2020. Therefore, it would be too complex to further divide the leaf cohorts at the phytomer (the dimension = 4 cohorts × 40 phytomers = 160). Since the photosynthesis was calculated at the PFT-level using the sum of all leaf biomass in each leaf age cohort, and the leaf biomass is relatively small compared to the branches and fruit biomass, we did not specifically simulate the leaf component at phytomer level but implemented the leaf as a whole with 4 leaf age cohorts at PFT level as a

simplification. We declared it in Section 2.3.1, Lines 150-152 when introducing the sub-PFT phenology ("In the model, only branches and fruit bunches were specifically simulated at each phytomer while leaf was simulated as a whole of all phytomers at the PFT level to reconcile with the four leaf age cohorts"). We also added the clarification in the caption of Figure 2 and modified several statements which will lead to misunderstanding in the manuscript.

For the leaf shedding, "the leaf longevity used in the VPD triggered leaf shedding scheme (eq. 2 and 3 in Chen et al., 2020) is modified to be the same than phytomer longevity (640 days) to approximate the old leaves removing in phytomers (it means than when all the 'leaves' dies, the phytomer dies). The shedding leaf then enters to the litter pool", This is added in Section 2.3.2, Lines 198-199 to explain this point. Considering that the removal of leaves is not very well represented at the time of phytomer pruning, we further added an extra leaf turnover of the old leaves (using the leaf age cohort) at the time when the oldest phytomer is manually pruned. We added this on Section 2.4.4, Lines 322-324.

**[Comment 13]** *L188: "a fixed allocation of 10% to reproductive plant tissues" – how this fraction of reproductive allocation is related to the phytomer specific fruit allocation? Fruit is usually considered part of the reproductive organ.*

**[Response to #13]** A fixed allocation of 10% to reproductive plant tissues is used in the original ORCHIDEE-MICT version for tropical evergreen forests. In ORCHIDEE-MICT-OP, the allocation to each fruit and branch of phytomer was not fixed and calculated as a fraction of the aboveground sapwood and reproductive organ using eq. 1. We deleted this sentence "A fixed allocation of 10% to reproductive plant tissues" to avoid the misunderstanding.

**[Comment 14]** *L198: according to the calibrated values of coefficients $P_1$, $P_2$ and $P_3$, the maximum value of Eq. (1) is $f_{br+fr,min} + (f_{br+fr,max} - f_{br+fr,min}) \times 0.07$ when a phytomer reaches max age; the range of the modifier 0 to 0.07 (instead of 0 to 1 normally) seems to be too small, which would exert very weak phenological effect on this allocation parameter.*

**[Response to #14]** We added a sentence on Section 2.3.3, Lines 217-218. to clarify this point: "Note that the modifier $f_{br+fr}^{i,nphs}$ range (0~0.07) is for one phytomer, and the total allocation fraction (a range of 0~1) should be the sum of modifiers in all phytomers."

**[Comment 15]** *Section 2.3.3: Inconsistency between model schematic in Figure 2 and allocation descriptions. In Figure 2, fruit to fruit-bunch allocation is independent of sapwood to branch allocation, but the descriptions and equations here suggest fruit allocation is part of sapwood allocation. Figure 2 also shows leaf to phytomer-leaf allocation, but there is no indication of this sub-PFT leaf allocation process in the text. Only a PFT- level allocation parameter $f_{leaf}$ is described in Eq. (5). If phytomer specific leaf carbon pool is not modeled, some parts of Figure 2 should be modified. And the text description of sub-PFT phenology for branch, leaf and fruit for each phytomer is not accurate (e.g., L28 in abstract).*

**[Response to #15]** The branch and fruit allocation are a part of aboveground sapwood and reproductive organ (fruit) allocation. The former is added to the sapwood pool, which goes to the litter pool after pruning, while the fruit allocation is added in the fruit pool, which is regularly harvested. We modified the inaccurate texts in Section 2.3.3., Lines 205-206 "The allocation to fruit and branch sub-component for each phytomer was calculated as a fraction of the aboveground sapwood and the reproductive organ".

In Figure 2, the allocation to leaf component is in dashed lines, and the branch and fruit component are presented in solid lines to indicate the sub-PFT level modeling of branch and fruit components without a specific leaf component. We also added the clarification in the caption of Figure 2, "The branch and fruit components (solid lines) were implemented at the phytomer level, while the leaf component (dashed lines) was not specifically simulated for each phytomer. The leaf LAI and biomass was implemented at the PFT level with four leaf age cohorts as a substitution" and modified statements in the Abstract (Lines 26-28) and Introduction (Section 1, Lines 98-100) to avoid misunderstanding.

**[Comment 16]** *L219: "fruits will be harvested after the phytomer age in the oldest phytomer reaches $Age_{ffbcrit}$–does this imply that fruit initiation happens at the same time as phytomer initiation? It needs to note that phytomers develop their leaves first before initiation of fruit. But the equations (1) to (5) suggest that fruit allocation at each phytomer starts immediately after phytomer initiation.*

**[Response to #16]** Yes, the phytomer starts the branch and leaf earlier and then the fruit. Here, we added a pre-defined variable "*ffblagday*", and "The initiation of fruit begins when the phytomer age exceeds the pre-defined *ffblagday* (16 days)." (Section 2.3.3, Lines 228-229). We also did a test to see whether the change of *ffblagday* (by adding or subtracting 5, 10 or 20% to the baseline values) leads to increase/decrease of the cumulative yield. As a result, the change of the *ffblagday* has a limited impact on the final cumulative yield (<0.01%) because the amount of the allocation to fruit in the early phytomer age is relatively low (see reply to #17).

**[Comment 17]** *Section 2.4: this section lacks a sensitivity analysis for newly introduced oil palm parameters. Although the results show general agreement with observational data by a one-time calibration with all sites, readers gain little insight on how sensitive are the oil palm LAI or biomass pools to different parameters and to climate and surface forcing without showing a sensitivity analysis or independent validation using different sites.*

**[Response to #17]** As suggested, we added a set of sensitivity test simulations to analyze the model sensitivities to the main parameters. We added section "*2.4.5 Sensitivity analysis*" in the Methods and section "*3.6 Sensitivity analysis*" in the Results.

2.4.5 Sensitivity tests

Because of the distinct age cohorts of oil palm and age-based parameterizations for photosynthesis and allocation in ORCHIDEE-MICT-OP, performing the sensitivity analysis on every age-specific parameter would be too CPU intensive. Instead, we performed sensitivity tests of the major parameters related to oil palm photosynthesis and allocation, particularly for the phytomer related allocation parameters without enough constraints from field observations. For the age-specific parameters (e.g., $V_{cmax25}$, *sla*), the calibrated value for CFT5 (the most productive phase with the maximum yield) were tested. The sensitivity tests were conducted by changing the selected parameters (variables with * in Table S2) by ±5, ±10 and ±20% from the originally calibrated value while keeping the other parameters unchanged. Their impacts on the cumulative yields at the most productive phase aging from 10-25 (corresponded to CFT5) were evaluated. For the grouped parameters such as the phytomer allocation coefficient ($P_1/P_2/P_3$), the sensitivity was tested by changing ±5, ±10 and ±20% of the target function ($F_{br+fr}^{i,nphs}$) using different combinations of P1~P3.

3.6 Sensitivity analysis results

The maximum rate of carboxylation ($V_{cmax25}$) is the most sensitive photosynthesis parameter because it determined the photosynthesis rates of leaf, followed by *sla*. Changes in $\pm 20\%$ of the baseline value of $V_{cmax25}$ leads to 13.8%/20.5% increase/decrease in the cumulative yields from age 10 to 25 (Figure 11). Maximum leaf area index ($LAI_{max}$), a threshold beyond which there is no allocation of biomass to leaves, has a smaller influence on the yields than $V_{cmax25}$ and *sla*. Yields are not changed linearly with changes in the $LAI_{max}$ value since it is a threshold parameter by definition.

For the allocation parameters, the empirical coefficients for the leaf ($L_1/L_2/L_3$) (Eq. 8) and root ($R_1$) (Eq. 9) allocation have very small impact on the fruit yields. The other allocation parameters are more or less related to the NPP allocation to aboveground sapwood and the reproductive pool, which influence the dynamics of the phytomer biomass and fruit yields. Among these parameters, yields are most sensitive to the phytomer allocation coefficients ($P_1/P_2/P_3$) (Eq. 1 and 2) which determine the NPP partitioning to phytomer (10% decrease in ($P_1/P_2/P_3$) leads to a decline of 21.23% in yield). The $f_{sab+rep,max}$ parameter controls the upper boundary of allocation to the aboveground sapwood and the reproductive organ (Eq. 11) and brings 19.4% increase in yields by changing +20% of the default value. Similarly, increasing/decreasing (10%) maximum fresh fruit bunch allocation fraction ($f_{fr,max}$) results in a significant increase/decrease (10%) of yields. By

contrast, changing the baseline values of $f_{sab+rep,min}$ , $f_{fr,min}$, $F_1$ (fruit bunch allocation coefficient), $\theta$ (the coefficient of partitioning allocation between above and belowground sapwood) and *ffblagday* leads to little influence on the final cumulative yields. The turnover-related parameter $LO_1$ exerts a negative impact on cumulative yields. The increase of $LO_1$ increased the old leaf loss throughout phytomer pruning and results in lower yield.

[Figure]

**Figure 11. Change in cumulative yields by varying ±5, ±10 and ±20% of the key parameters related to photosynthesis, allocation and turnover in the oil palm modelling. The parameter is changed one by one while the others are kept as the same.**

[**Comment 18**] *L227: missing Yin and Struik, 2009 in the reference list.*

[**Response to #18**] Thanks for the reminding. We added it to the reference list.

*Yin, X., and Struik, P. C.: C3 and C4 photosynthesis models: An overview from the perspective of crop modelling, NJAS - Wageningen Journal of Life Sciences, 57, 27-38, https://doi.org/10.1016/j.njas.2009.07.001, 2009.*

[**Comment 19**] *Section 2.4.3: I suggest merging this section to section 2.3.3 as they both describe allocation. Again, leaf allocation ($f_{leaf}$) is only described for the whole palm, not for each phytomer. If only an PFT integrated leaf carbon pool is grown for the whole palm, Figure 2 and several places in the text about sub-PFT structure should be modified.*

[**Response to #19**] Section 2.3 described the implementation of the oil palm phytomer including the Section 2.3.1 introduction of the phytomer structure, Section 2.3.2 phytomer phenology, Section 2.3.3 allocation and Section 2.3.4 the regular fruit harvest. While Section 2.4.3 is the recalibration of parameters in the existing equations mainly based on field observations and previous literature. We understand it is more connected by the contents of these two parts (both for allocation), but Section 2.3.3 is the sub-PFT level allocation while Section 2.4.3 exhibits the PFT-level allocation. We believe the separation between model development and parameterization would be more helpful to concentrate on the phytomer level modification from the original model in Section 2.3. It will also help the readers to better differentiate the differences in the PFT-level allocation in leaf and phytomer-level allocation of branch and fruit. We changed the title of Section 2.3.3 from allocation to Section 2.3.3 phytomer allocation to discriminate these two sections.

For the leaf allocation, we declared it in  when introducing the sub-PFT phenology. We also added clarification in the caption of Figure 2 and modified sentences in the abstract to avoid misunderstanding.

**[Comment 20]** *Section 2.4.4: Since Ageleafcrit equals Agephycrit, one of these parameters could be eliminated. But it needs to note that, in reality the **leaf longevity is smaller than phytomer longevity** as there is a lengthy "spear leaf" (a bud that grows to as long as ~3 meters) stage before the it fully expands to be able to photosynthesize.*

**[Response to #20]** We deleted the duplicated $Age_{leafcrit}$ as suggested. In our model, we simplified the leaf growth without considering a "spear leaf" stage. This could be done in the future. We also ran a test simulation using a shorter $Age_{leafcrit}$ (Test1, $Age_{leafcrit} = 620 < Age_{phycrit} = 640$) (see the Figure below). The decreased leaf longevity accelerates leaf shedding and causes a compensatory increase in leaf allocation. NPP and cumulative yields also increased because of the increase of new leaf proportion with higher photosynthesis capacity. We added the figure in the revised version (Figure S8) and sentences on

[Figure]

**Figure S8. Changes in the simulated variables using different settings for longevity and shedding. 1) using the leaf longevity (620 days) shorter than phytomer longevity (640 days) and 2) turn off the extra old leaf turnover at the time of oldest phytomer pruning.**

**[Comment 21]** *L300: if leaf carbon pool is not specifically simulated for each phytomer like branches and fruits, how pruning of each leaf is conducted, and how this is reconciled with the VPD-triggered leaf shedding? It needs to avoid double accounting of leaf litterfall flux from these two processes. In oil palm plantations, natural leaf loss/shedding is almost impossible without pruning due to very high lignin content. Thus, I suggest only one leaf litterfall mechanism is implemented for oil palm.*

**[Response to #21]** For the details of the implementation of leaf and the leaf shedding scheme in our model, please see the reply to #12. We also did a test (Test 2, turn off the extra old leaf turnover) according to the suggestion by removing the extra leaf turnover when the phytomer is manually pruned. The results showed a decrease in the simulated GPP and biomass pool but an increase in NPP and cumulative yields (Figure S8, test 2 in reply to 19#).

**[Comment 22]** *L350-355: it will be more convincing to compare the simulated annual yield with real observations rather than the fitted curve in Figure 7d. This can be achieved either from using continuous harvest data or using space-for-time substitution with multiple sites.*

**[Response to #22]** Yes, we agree with the reviewer. However, it is difficult to obtain enough annual yield observations from the public available data, and most published studies provided yield records at one time phase. Therefore, we used the only harvest records that we can access in Figure 7d. Facing the difficulty in acquiring the original harvest records for independent sites, we also ran simulations in two independent sites

(Teh and Cheah 2018 and Fan et al., 2015) and visually compare the temporal dynamics of simulated yields. For the details of the comparison and the supplement figure, please see the reply to #8.

**[Comment 23]** *L376-377: calibration an LSM using all sites without reserving some sites for inde- pendent validation is not standard. Also, the reason of overestimation should not be attributed to "no calibration was applied for this site". LSM is aimed for regional or global applications. Thus, even a perfect calibration for an individual site does not mean good predictive power for other sites. That's why I urge the authors to do independent vali- dation after model calibration using separate datasets.*

**[Response to #23]** Please see the reply to #8.

**[Comment 24]** *Section 3.5: again, the fact that leaf LAI and biomass pool is not specifically simu- lated for each phytomer should be clarified much earlier when introducing the sub-PFT structure and in Figure 2.*

**[Response to #24]** We declared "In the model, only branches and fruit bunch were specifically simulated at each phytomer while leaf was simulated as a whole of all phytomers at the PFT level to remain consistent with the four leaf age cohorts of the modelled phenological equations" in Section 2.3.1, Lines 150-152 when introducing the sub-PFT structure. We also added clarification in the caption of Figure 2. Please also refer to the reply to #12 and #15.

**[Comment 25]** *L412: "The fruit production and harvest begin after entering the fruit development phase" can be described earlier when introducing Figure 3. The 3-phase description is confusing as it sounds like only the productive phase (CFT5) has yield and harvest.*

**[Response to #25]** We modified and added sentences when introducing the 3-phase in Section 2.3.2, Lines 168-171, "The second phase is the fruit development phase when fruit begins to grow and harvest," and Figure 3 (for the details, please refer to reply to #9).

**[Comment 26]** *L440-441: Fan et al. implemented age-dependent carbon allocation strategy (their section 2.2.1) and showed age-dependent trend in yields validated against different sites (their Figure 10).*

**[Response to #26]** Here, we wanted to state that the distinct age classes of oil palm and the age-specific parameterization scheme is new for oil palm in LSMs, not only the age-dependent carbon allocation strategy. We modified the text to "To our best knowledge, distinct age classes of oil palm and the age-based parameterizations for photosynthesis and autotrophic respiration dynamics have not yet been implemented in the previous LSMs aiming to simulate oil palm biophysical variables." Section 4.1, Lines 497-499.

**[Comment 27]** *L445: this is not true; Fan et al. used 2 sites for model calibration and an additional 8 sites for independent validation. The study here used all 14 sites for both calibration and validation, which violates the normal procedure of model validation using independent sites and undermines its generalizability to other locations.*

**[Response to #27]** Please see reply to #8 for the details about calibration and validation. For clarification, we modified this sentence to "Moreover, the calibration for age-specific parameters is based on the 14 observation sites with variable climate and soil conditions, and we also compared the simulation results with observations for a range of variables including biomass, yield, LAI, GPP and NPP and biomass/GPP component." Section 4.1, Lines 501-505.

**[Comment 28]** *L446: this is not necessarily the case; parameterizations from ORCHIDEE-MICT-AP can only be applicable to other LSMs if they share very similar model structures. Without a sensitivity analysis and independent validation, such claim is premature.*

**[Response to #28]** We agree with the reviewer. We added the sensitivity analysis in Section 3.6, Lines 464-480 and tried independent validation (see the reply to #8). We also modified this sentence to "Therefore, our

parameterizations of oil palm (Table S2) can also be a reference as for other LSMs." Section 4.1, Liotnes 504-505.

[**Comment 29**] *Section 4.3: when talking about application in LUC impact assessment, it is important to note the limitation that effects of different land cover types, soil carbon and nutri- ent content, and fertilization management on oil palm growth and yield can hardly be represented by the current model without a nitrogen cycle.*

[**Response to #29**] We added the limitations in Section 4.3 as suggested. "…, although the effects of soil carbon and nutrient content, and fertilization management on oil palm growth and yields still require further investigation." Section 4.3, the last sentence.

---

## Author Comment (AC2) · 5 Apr 2021

*Referee #2*

**[General comment]** *This paper presents an improved, oil palm specific plant functional type (PFT) to be incorporated into the ORCHIDEE-MICT global land surface model. The authors offer this as an improvement on the existing typical practice of approximation using the PFT of tropical broadleaved evergreen trees. Given the intensively managed, and specific, nature of palm cultivation and harvesting there is little doubt as to how unsatisfactory the utilisation of a generic plant functional type is in modelling physiological and environmental dynamics in these systems. The paper is generally very well written and presented (with some caveats, see details below) and the results suggest a significant improvement over previous model practice. Reviewer #1 has provided some excellent and comprehensive technical comments and suggestions and I have little to add to those. I would comment though, that peat and mineral soil based plantations are likely to be very different, yields will be lower and palm mortality higher on tropical peat soils and crop rotations significantly shorter (18 to 20 rather than 25 to 30 years). With around 25% of the South East Asian oil palm plantations being on converted peatlands, a discussion around the significant differences likely to be found in yield and palm mortality (disease incidence, palm root anchorage and failure) on the two soil types and whether they should be separately parameterised, and what work would be required to do so, in the PFT would be welcome.*

**[Response]** We thank the reviewer for the careful review and great suggestions on the discrimination of peat and mineral based oil palm plantations. Accordingly, we modified several inaccurate statements in the manuscript (please see the reply to #6, #12, #23). We also added discussion about oil palm on the peat soil and mineral soil and the potential improvement regarding the inclusion of these two soil types in our future work in Section 4.2, Lines 550-567, "Another important factor is the difference between oil palms grown on mineral and peat soils. Although our model was able to reproduce the yield, GPP and NPP at one peat-based oil palm site (Site 12), the biomass is overestimated throughout the life cycle, indicating further work is needed to implement the peat oil palm in the LSMs (and other data from peat soils for yields). Previous studies suggested that the frond biomass of oil palm grown on peat soils was lower than on mineral soils in all age classes (Henson 2005). On peat soil, oil palm allocates less biomass to root system (Corley, Gray and Kee 1971; Othman et al., 2010). Further decomposition of peat subsidence after peatland drainage combined with poor anchorage of oil palm may cause palm leaning and even palm falling and hence increase mortality (Henson et al., 2003; Othman et al., 2010). Based on the yield and tree mortality, the rotation cycle also varies in mineral- (25-30 years) and peat- (18-20 years) based oil palm. A better representation of peat oil palm could be reached by using a separate parameterization scheme for peat oil palm (e.g., adjusting the partition between AGB and BGB and decrease the carbon assimilation rate), adopting a lower biomass threshold for oil palm rotation (Figure 5), modifying the carbon emission rate at the beginning years of oil palm conversion and so on. However, it would be a great challenge to implement some factors such as disease in the current stage without enough knowledge on the processes and impacts of disease on oil palm growth. Also, we note the optimal planting density is different between the two soil types (110-148 palms ha$^{-1}$ on mineral soil and 160-200 palms ha$^{-1}$ on peat soil) (Henson et al., 2003; Othman et al., 2010; Lewis et al., 2020). The mineral-based oil palm suffers a decline in frond biomass and production while that of the peat oil palm is less influenced (Lewis et al., 2020). These would also cause biases in simulated biomass and yield due to no separation between mineral- and peat-based oil palm." One ORCHIDEE version— ORCHIDEE-PEAT —already implemented the peat process in high latitudes (Qiu et al., 2018). Merging the oil palm specific morphology, phenology and harvest process of oil palm and the peat related process between these two ORCHIDEE versions would be our next step to better simulate oil palm yields and carbon, water and energy fluxes on peat soils. We added this point in Section 4.3, Lines 580-584). For the details of the discussion, please see the reply to #12, #15, #22, #23. Besides, we also added some explanations in Figure 3 to make it clear.

All of the specific comments and suggestions have been addressed and implemented in this revised manuscript. Please find below the specific reviewer's comments, followed by our responses and relevant changes in the manuscript. We believe that the revised version addresses all the issues raised by the reviewer.

**Specific comments:**

**[Comment 1]** *ln.20: "cause" should be "causes" (plural)*

**[Response to #1]** Thanks for the suggestion. We changed "cause" to "causes".

**[Comment 2]** *ln.36: "crop" should be "crops" (plural)*

**[Response to #2]** We modified "crop" to "crops".

**[Comment 3]** *ln.48: citations in the brackets should be in chronological order*

**[Response to #3]** We modified the orders of the citations as suggested (please see Section 1, Lines 47-48, 67-68, 268-269, 317, 546-547, 587).

**[Comment 4]** *ln.54-55: "do not allow to represent the land use change", poor English, perhaps: "do not allow representation of land-use change..."*

**[Response to #4]** We modified this sentence as suggested.

**[Comment 5]** *ln.60-61: change "according to the genotypes and locations" to "dependent on genotype and locations..."*

**[Response to #5]** We changed this sentence as suggested.

**[Comment 6]** *ln.84: rotations of 25-30 years would be specific to mineral soil-based plantations, peat-based plantations are likely to be significantly shorter*

**[Response to #6]** Thanks for your suggestions. We changed this sentence to "Also, oil palm planted in mineral soil is managed in a rotation cycle of 25-30 years (manually cut) due to the difficulties in harvesting and the potential decline of fruit production (Hoffmann et al., 2014; Röll et al., 2015)" (Please see Section 1, Lines 84-86) and discussed the rotation cycle of peat land oil palm in the discussion part (Section 4.2, Lines 557-558).

**[Comment 7]** *ln.148: change "corresponded" to "corresponding"*

**[Response to #7]** Revised accordingly.

**[Comment 8]** *ln. 158: "begin to flourish"*

**[Response to #8]** We modified this sentence as suggested.

**[Comment 9]** *ln.201: a space needed "accelerated (Corley and Tinker, 2015)*

**[Response to #9]** Thanks for the reminding. We added a space between accelerated and (Corley and Tinker, 2015).

**[Comment 10]** *ln. 222: delete "types", already included in the PFT acronym*

**[Response to #10]** We deleted "types".

**[Comment 11]** *ln.241: change "correspondingly" to "corresponding"*

**[Response to #11]** We modified "correspondingly" to "corresponding".

**[Comment 12]** *ln.305: "we assumed that there is no natural mortality for the oil palm,", this is un- realistic as there will be disease losses from the original planting density (with some replacement) and specifically in peatlands, losses and yield reduction due to leaning palms. These need discussing.*

**[Response to #12]** We changed this sentence to "In ORCHIDEE-MICT-OP, we assumed that oil palm is manually cut down for rotation before the natural mortality without considering the disease and other causes of tree loss as well" (Please see Section 2.4.4, Lines 330-332). Disease was not considered in our model since it is difficult to quantify the impacts of diseases on oil palm in a grid-based land surface model. Implementing the individual tree disturbance for PFT-level oil palm simulation would be a great challenge and we also admit the limitation in the Discussion (Please see Section 2.4.4, Lines 558-563) "A better representation of peat oil palm could be reached by using a separate parameterization scheme for peat oil palm (e.g., adjusting the partition between AGB and BGB and decrease the carbon assimilation rate), adopting a lower biomass threshold for oil palm rotation (Figure 5), modifying the carbon emission rate at the beginning years of oil palm conversion and so on. However, it would be a great challenge to implement some factors such as disease in the current stage without enough knowledge on the processes and impacts of disease on oil palm growth.".

**[Comment 13]** *ln.313: "Figure S3" should be "Figure 3"*

**[Response to #13]** Sorry for the mistake. It was corrected in the revised version.

**[Comment 14]** *ln.369: "is reproduced of 1.7" Perhaps this should be "is calculated at..."?*

**[Response to #14]** Yes, we modified this sentence as suggested.

**[Comment 15]** *ln.376: this distinction between peat and mineral-based plantations, and its implication for scaling model output needs to be discussed further*

**[Response to #15]** As suggested, we further discussed the distinction between peat and mineral-based plantations and implications for model parameterization in Section 4.2. Lines 550-563 "Another important factor is the difference between oil palms grown on mineral and peat soils. Although our model generally was able to reproduce the yield, GPP and NPP at one peat-based oil palm site (Site 12), the biomass is overestimated throughout the life cycle, indicating further work is needed to implement the peat oil palm in the LSMs (and other data from peat soils for yields). Previous studies suggested that the frond biomass of oil palm grown on peat soils was lower than on mineral soils in all age classes (Henson 2005). On peat soil, oil palm allocates less biomass to root system (Corley, Gray and Kee 1971; Othman et al., 2010). Further decomposition of peat subsidence after peatland drainage combined with poor anchorage of oil palm may cause palm leaning and even palm falling and hence increase mortality (Henson et al., 2003; Othman et al., 2010). Based on the yield and tree mortality, the rotation cycle also varies in mineral- (25-30 years) and peat- (18-20 years) based oil palm. A better representation of peat oil palm could be reached by using a separate parameterization scheme for peat oil palm (e.g., adjusting partition between AGB and BGB and decrease the carbon assimilation rate), adopting a lower biomass threshold for oil palm rotation (Figure 5), modifying the carbon emission rate at the beginning years of oil palm conversion and so on. However, it would be a great challenge to implement some factors such as disease in the current stage without enough knowledge on the processes and impacts of disease on oil palm growth.

Henson, I. E., and Dolmat, M. T.: Physiological analysis of an oil palm density trial on a peat soil, *Journal of Oil Palm Research*, 15, 2003.

Henson, I. E.. Modelling vegetative dry matter production of oil palm. *Oil Palm Bulletin*, 52, 25, 2005

Corley, R. H. V., Gray, B. S., & Kee, N. S. (1971). Productivity of the oil palm (Elaeis guineensis Jacq.) in Malaysia. *Experimental Agriculture*, 7(2), 129-136.

Lewis, K., Rumpang, E., Kho, L. K., McCalmont, J., Teh, Y. A., Gallego-Sala, A., and Hill, T. C.: An assessment of oil palm plantation aboveground biomass stocks on tropical peat using destructive and non-destructive methods, *Scientific Reports*, 10, 2230, 10.1038/s41598-020-58982-9, 2020.

Othman, H. A. S. N. O. L., Mohammed, A. T., Harun, M. H., Darus, F. M., & Mos, H. A. S. I. M. A. H. , *Best management practises for oil palm planting on peat: optimum groundwater table. MPOB Information Series, 528, 1-7, 2010.*

**[Comment 16]** *ln.408: change "characteristics" to "characteristic" (singular)*

**[Response to #16]** We modified "characteristics" to "characteristic".

**[Comment 17]** *ln.436: change "special" to "specific"*

**[Response to #17]** We changed "special" to "specific".

**[Comment 18]** *ln.440: change "increasing" to "increase"*

**[Response to #18]** We changed "increasing" to "increase".

**[Comment 19]** *ln.440-441: "To our best knowledge, age-based allocation dynamics for oil palm have not yet been..."*

**[Response to #19]** We modified this sentence as suggested.

**[Comment 20]** *ln.441: change "simulating" to "simulate"*

**[Response to #20]** The word "simulating" was modified to "simulate".

**[Comment 21]** *ln.452: change "phytomer" to "phytomers" (plural)*

**[Response to #21]** We changed "phytomer" to "phytomers".

**[Comment 22]** *ln.487-488: the authors mention the impact that planting density will have when scaling from palm to area, again this will differ between peatland and mineral and needs more discussion.*

**[Response to #22]** We added the discussion of the influence of planting density on peatland and mineral soils in Section 4.2 Lines 563-567, "Also, we note the optimal planting density is different between the two soil types (110-148 palms ha$^{-1}$ on mineral soil and 160-200 palms ha$^{-1}$ on peat soil) (Henson et al., 2003; Othman et al., 2010; Lewis et al., 2020). The mineral-based oil palm suffers a decline in frond biomass and production while that of the peat oil palm is less influenced (Lewis et al., 2020). These would also cause biases in simulated biomass and yield due to no separation between mineral- and peat-based oil palm".

**[Comment 23]** *ln.496-497: The authors state that "soil carbon change may take a long time", they do not elaborate on what a "long time" is, but in peat systems there are huge carbon emissions observed in the first 5 years following conversion, likely a similar amount of time needed for forest residues to decompose which would disagree with the thrust of this sentence.*

**[Response to #23]** Thanks for the great point. We modified this sentence to "In reality, the biomass loss from deforestation is fast but soil carbon change may take a long time in mineral soil. A more complex condition would happen in the conversion to oil palm plantation on the peat soil, where huge carbon emission was observed in the first 5 years following conversion (Hooijer et al., 2012; Cooper et al., 2020)" Section 4.3, Lines 575-578. We also discussed the potential implementation of the conversion from peat soil to oil palm plantation based on ORCHIDEE-MICT-OP and another branch of ORCHIDEE for high-latitude peatlands (ORCHIDEE-PEAT) in Section 4.3, Lines 580-583. "One of the ORCHIDEE branches, ORCHIDEE-PEAT, has already implemented the peat processes for high latitudes (Qiu et al., 2018). Merging the oil palm specific morphology, phenology and harvest processes of oil palm and the peat related

processes in these two branches would help characterize the oil palm yields as well as carbon, water and energy fluxes on peat soil palms."

Cooper, H. V., Evers, S., Aplin, P., Crout, N., Dahalan, M. P. B., and Sjogersten, S.: Greenhouse gas emissions resulting from conversion of peat swamp forest to oil palm plantation, Nat. Commun., 11, 407, 10.1038/s41467-020-14298-w, 2020.

Hooijer, A., Page, S., Jauhiainen, J., Lee, W. A., Lu, X. X., Idris, A., & Anshari, G. Subsidence and carbon loss in drained tropical peatlands. Biogeosciences, 2012,9(3), 1053-1071.

Qiu C, Zhu D, Ciais P, Guenet B, Krinner G, Peng S, et al. ORCHIDEE-PEAT (revision 4596), a model for northern peatland CO2, water, and energy fluxes on daily to annual scales. Geosci Model Dev.11(2):497-519, 2018.

[**Comment 24**] *ln. 499-500: change ",it is also in urgency to..." to ",there is an urgent need to..." ln.504: change "crops" to "crop" (singlular)*

[**Response to #24**] We modified these sentences as suggested.

[**Comment 25**] *Figure 3: change "prunning" to "pruning". Also, please include full versions of acronyms used in the figures in their associated legends (e.g. CFT in fig.3). The legend for fig. 3 is generally inadequate and needs more detail. It should stand alone and not require recourse back to the text.*

[**Response to #25**] The word "prunning" was changed to "pruning". We added the full versions of the acronyms in Figure 2 and Figure 3. We also added the details in Figure 3 and the captions. Please see the figure below.

[Figure]

**Figure 3 Schematic of (a) leaf, (b) phytomer and (c) plant dynamics with leaf, phytomer and tree ages. The branch and fruit allocation is a function of phytomer age. The oil palm PFT experiences an increase of fruit yield during CFT 2-4 and reaches the maximum and steady yield at the most productive period (CFT5). The leaf component is not specifically simulated for each phytomer (dashed rectangle) but implemented at the PFT level with four leaf age cohorts. The major phenological phases for phytomer during the oil palm life cycle are presented with tree ages. LC and CFT refer to leaf cohort and cohort functional type, respectively.**

---

## Author Response (AR2)

*Referee #1*

**[Comment]**

The authors carefully addressed the comments and largely clarified the technical details of model development and eliminated some inaccurate statements in revising the manuscript. The added sensitivity analysis and visual comparison to some published yield dynamics are very helpful. Overall, the paper is much improved.

Yet, there is still one key component of the new oil palm model, that is, leaf phenology (initiation and shedding scheme), requiring additional care.

First, just as the authors mentioned natural tree mortality is not applicable for oil palms (L330), the natural leaf shedding scheme implemented for TBE trees in ORCHIDEE from Chen et al. (2020) also does not really apply to oil palms. Trees have numerous small leaves with various timing of initiating and shedding throughout the canopy. But palm's canopy is highly stratified and the ~40 big leaves and their leaflets are highly lignified, which do not shed naturally even after full senescence but are removed by plantation managers with regular pruning at the bottom layer. The big leaves each grows 1.5-2 years in sequence from the top to bottom (note the leaflets of each big leaf are of the same age and phenology). More details about oil palm's monopodial morphology and its difference from tree phenology can be found in Corley and Tinker (2015). Therefore, the leaf shedding scheme of four leaf age cohorts (eq. 2 and 3 in Chen et al.) that pertains to natural trees may not apply to oil palm. It is also not convincing that the instantaneous meteorological drivers (like weekly VPD, L196-198) could drive oil palm's canopy phenology given the long-term development of each big leaf. Instead, the extra leaf loss mechanism (L322-326) at the time when the oldest phytomer is manually pruned (Eq. 10) could be the more proper leaf shedding scheme for oil palm than the Chen et al. (2020) scheme. So in Figure S8, why not test excluding the VPD-triggered leaf shedding mechanism but using only the pruning mechanism (oldest phytomer) for oil palm? That's why I commented about avoiding double accounting of leaf litterfall flux especially when VPD-triggered leaf shedding may not exist for oil palm in the real world.

Similarly, the assumption that new leaf allocation (fleaf) is related to SWdown and the light transmission of old leaves (eq. 1 in Chen et al.) may not be suitable for oil palm, because palm's old leaves are always at the bottom of canopy and do not affect light availability to new leaves which are always at the top. So the authors should at least note in L196-197 (track change file) and/or in discussion the limitation of this leaf allocation mechanism for oil palm.

Although the current model can reasonably capture the multi-site LAI, GPP, NPP, biomass and yield dynamics, it does not yet represent oil palm's unique canopy phenology which governs canopy photosynthesis and transpiration and related water and energy fluxes, though leaf carbon pool is small, and may affect other aspects of the LSM. Since the paper lacks calibration and/or validation on simulated leaf litterfall, it is hard to tell whether the leaf phenology and litterfall flux simulated with the Chen et al. (2020) mechanism matches observations. Parsimonious model development is important, but when some unsuitable mechanisms from trees are used for palms, there may arise additional uncertainties in modeled fluxes at the diurnal, seasonal, or interannual time scales (especially if this model will be evaluated with eddy covariance fluxes in the future). The distinct age classes of oil palm and the age-based

parameterizations for photosynthesis and autotrophic respiration dynamics is an important advance in this model, and this feature could perhaps be utilized without including the whole leaf phenology schemes that are designed for nature trees. In sum, the VPD and SWdown triggered leaf shedding and initiation schemes from Chen et al. should be reconsidered for its applicability to oil palm, given its distinction from "trees".

**[Response]** We thank the reviewer for the careful review again and suggestions on the leaf initiation and shedding scheme. In fact, based on the reviewer's comment in the last round, we already separated the two leaf shedding schemes as independent modules. Thus, it is flexible for the users to activate either scheme.

Accordingly, we added discussion about uncertainty and limitation of the implemented leaf phenology in Section 4.2, Lines 525-533, "In our model, we adopted the leaf phenology scheme from Chen et al (2020), which is preliminarily developed for tropical forests. We also added an extra old leaf turnover at the time of oldest phytomer pruning according to the regular management practice of phytomer pruning. However, whether the leaf initiation and leaf shedding schemes are suitable for oil palm requires further investigation, and more field evidence and control experiments are needed to reveal the mechanism of leaf shedding. Because of the limited understanding of oil palm leaf shedding mechanisms other than leaf removal along with phytomer pruning, these two leaf shedding schemes were both implemented in our model. Either or both schemes can be easily chosen using an external switch (pruning- or VPD-triggered leaf shedding scheme or combined). With more field observations become available in the future, the model is flexible to adapt the emergent mechanism, but some parameter calibrations may be needed."